# ProTransformer: Robustify Transformers via Plug-and-Play Paradigm

**Zhichao Hou**[1]     **Weizhi Gao**[1]     **Yuchen Shen**[2]     **Feiyi Wang**[3]     **Xiaorui Liu**[1] [*]

[1]North Carolina State University, [2]Carnegie Mellon University, [3]Oak Ridge National Laboratory

{zhou4,wgao23,xliu96}@ncsu.edu yuchens3@cs.cmu.edu fwang2@ornl.gov

## Abstract

Transformer-based architectures have dominated various areas of machine learning in recent years. In this paper, we introduce a novel robust attention mechanism designed to enhance the resilience of transformer-based architectures. Crucially, this technique can be integrated into existing transformers as a plug-and-play layer, improving their robustness without the need for additional training or fine-tuning. Through comprehensive experiments and ablation studies, we demonstrate that our ProTransformer significantly enhances the robustness of transformer models across a variety of prediction tasks, attack mechanisms, backbone architectures, and data domains. Notably, without further fine-tuning, the ProTransformer consistently improves the performance of vanilla transformers by 19.5%, 28.3%, 16.1%, and 11.4% for BERT, ALBERT, DistilBERT, and RoBERTa, respectively, under the classical TextFooler attack. Furthermore, ProTransformer shows promising resilience in large language models (LLMs) against prompting-based attacks, improving the performance of T5 and LLaMA by 24.8% and 17.8%, respectively, and enhancing Vicuna by an average of 10.4% against the Jailbreaking attack. Beyond the language domain, ProTransformer also demonstrates outstanding robustness in both vision and graph domains. Our code is available at https://github.com/chris-hzc/ProTransformer.

## 1 Introduction

In recent years, attention mechanisms and transformer-based architectures have drawn significant attention across many domains in machine learning, such as natural language processing (NLP) [1, 2], computer vision [3, 4], and graph learning [5, 6]. In particular, transformers have demonstrated superior capabilities to learn and model complex relations in data through powerful and universal attention mechanisms, and they have dominated many popular NLP tasks such as topic classification, sentiment analysis, textual entailment, machine translation, dialogue generation, etc [2]. Despite their success in NLP and beyond, many recent studies have demonstrated that transformers are highly vulnerable to adversarial attacks such that even small modifications to the input can easily fool the model [7, 8, 9]. However, most research on transformer architectures focuses on accuracy and efficiency, largely ignoring their security and robustness [10, 11].

With the increasing popularity of Large Language Models (LLMs) [12, 13], the robustness and security concerns of transformer architectures become particularly of interest. It has been shown that malicious attackers can invade the language models through various approaches as shown in Figure 1. The attacker can modify the input content in text attacks [14] or the prompt template in

---

[*]Corresponding author.

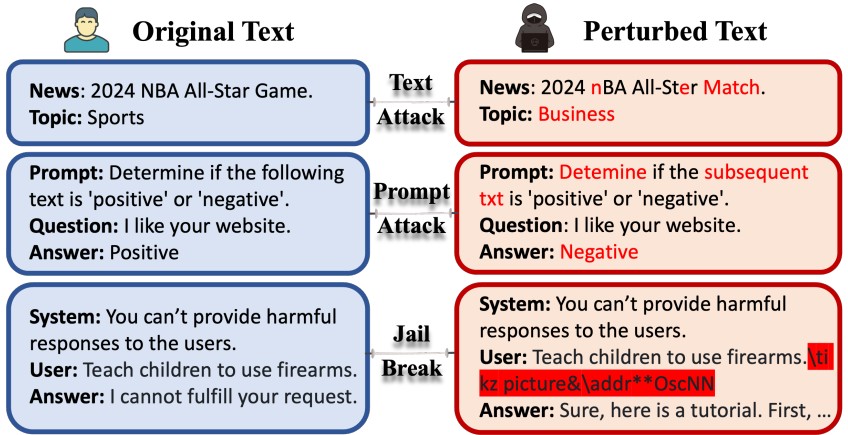

Figure 1: Various attack mechanisms on language models. *Classic text attacks* modify the input content using typos or synonyms; *Prompt attacks* perturb the prompt template within the input; and *Jailbreaks* append adversarial, non-semantic suffixes to manipulate the model into producing malicious outputs.

prompt attacks to mislead the model predictions [15]. Moreover, by adding adversarial suffixes, the jailbreaking attack [16] can prompt a LLM to generate toxic and illegal content which could lead to catastrophic legal and ethical impacts such as malicious speech or privacy leaks. Given the broad applications of transformers and their vulnerabilities under attacks, it is imperative to design a universal and effective strategy to enhance the robustness of transformers.

Existing research attempting to improve the robustness of transformers can be roughly divided into empirical defenses [17, 18, 19, 20, 21] and certifiable defenses [22, 23, 24, 25]. Nevertheless, these defenses require excessive computation costs for training, inference, or both. In addition to these architecture-agnostic defenses, there are also several works proposing to enhance the robustness of transformers architecture [26, 27, 28, 29]. However, these approaches either require substantial computations or rely on specific domain knowledge, which hinders their extensions to larger models or broader application domains.

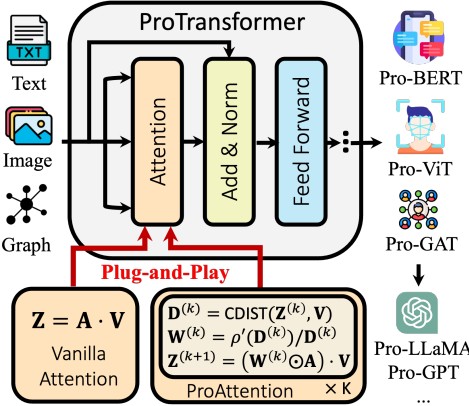

Figure 2: Overview of ProTransformer. ProAttention can be plugged into pretrained transformers without additional training. The ProTransformer is versatile and can be applied across various domains, including language, image, and graph.

In this paper, given the limitations of existing works and the enormous training cost of transformers, we aim to robustify transformer architectures via a plug-and-play paradigm without additional training or fine-tuning. Our proposed *ProAttention* (Algorithm 1) can be readily plugged into the given transformers to convert them to *ProTransformer* (as shown in Figure 2) with significantly stronger robustness. Specifically, our contributions can be summarized as follows:

- We establish a novel connection between the attention mechanism in transformers and the weighted least square estimator. We provide interpretation and numerical simulation to reveal its vulnerability against potential adversarial attacks.

- From our new perspective, we propose robust token estimators to improve the resilience of token aggregation against adversarial attacks. We also propose an efficient Newton-IRLS algorithm to approximate the non-convex and non-smooth robust token estimator with convergence guarantees. The derived algorithm can be plugged into the given transformer as a plug-and-play layer to enhance its robustness against attacks even without additional training or fine-tuning.

- Our comprehensive experiments and ablation studies demonstrate that the proposed ProTransformer is effective, efficient, and generalizable. It significantly improves the robustness of transformers across various machine learning tasks, attack mechanisms, backbone architectures, and data domains such as language, vision, and graphs.

## 2    Related Work

In this section, we mainly summarize related works on the attacks and defenses of transformers focusing on language domains since this is the focus of this paper.

**Attacks.** Compared to the attack mechanisms in vision domain [30, 31], the text attacks in the language domain are highly complicated due to the natural irregularity of data structure. According to the perturbation units, text attacks can be classified into character-level [7, 32], word-level [33, 34, 35, 36, 9, 14, 37], sentence-level [38], and multi-level [39, 40, 8]. These classic text attacks typically generate adversarial examples through misspellings, synonym replacement, etc. In the era of LLMs, several new types of attacks have emerged, such as jailbreak attacks [16, 41, 42, 43] and prompt injection [44, 45, 46]. These prompting-based attacks aim to trick models into generating unsafe outputs using adversarially crafted prompts.

**Defenses.** There have been some works proposed to defend against adversarial text attacks from various perspectives. Empirical defenses, such as data augmentation [17] and adversarial training [47, 18, 19, 20, 21], attempt to robustify models by exposing them to a wider range of adversaries during training. On the other hand, several certifiable defenses [24, 25, 22, 23] have been proposed to guarantee the model robustness regardless of the attacks. However, these defenses require excessive computation costs for training, inference, or both, which limits their application in large-scale problems such as LLMs. Besides, all these methods are typically architecture-agnostic, which are orthogonal to and can be combined with our proposed defenses on the transformer architecture to further enhance the robustness.

To safeguard the transformers, several endeavors have been made from the transformer architecture perspective. Li et al. [26] modify the attention mechanism and position embedding to robustify text-to-speech transformers. In the crisis detection and recognition task, Liu et al. [27] propose an end-to-end attention-based classifier to enhance robustness. For tabular data, TableFormer [28] adopts structural-aware table-text encodings that are more robust to row and column order perturbations. However, these architectures are tailored for specific tasks, which require specific domain knowledge and can not be generalized across tasks. Han et al. [29] propose a general framework for self-attention modules via robust kernel density estimation (RKDE). However, this method introduces excess computation cost and shows relatively limited robustness improvement. Generally speaking, existing approaches either require substantial computations or rely on specific domain knowledge, which hinders their extensions to larger models or broader application domains.

## 3    ProTransformer

The main goal of this paper is to design robust self-attention mechanisms that are more resilient to adversarial attacks so they can be applied to robustify Transformer architectures. In this section, we first provide a new interpretation of the self-attention mechanism in Transformer architecture as the weighted least-square token estimator in Section 3.1. Then we propose robust token estimators that are more resilient to the dominating impact of input tokens in Section 3.2. An efficient Newton-IRLS algorithm is derived with a convergence guarantee to approximate the robust token estimator in Section 3.3. Finally, we describe how the proposed algorithm can be unrolled as robust attention layers to enhance the robustness of transformer architectures in Section 3.4.

### 3.1    Attention Mechanism as WLS Token Estimator

First, we provide a new perspective to formulate the vanilla attention mechanism as the weighted least squares (WLS) token estimator. In the self-attention layer, each output token $\mathbf{z}$ aggregates the values of input tokens $\{\mathbf{v}_j\}$ as their weighted sum according to the attention weights: $\mathbf{z} = \sum_{j=1}^{N} a_j \mathbf{v}_j$, where $\{a_j\}_{j \in [N]}$ are the attention weights and $\{\mathbf{v}_j\}_{j \in [N]}$ are value vectors for all $N$ input tokens. This weighted sum can be interpreted as the optimal solution of the following weighted least squares

(WLS) error minimization problem:

$$\arg\min_{\mathbf{z}} \mathcal{L}(\mathbf{z}) = \sum_{j=1}^{N} a_j \cdot \|\mathbf{v}_j - \mathbf{z}\|^2, \tag{1}$$

whose first-order optimality condition ($\nabla\mathcal{L}(\mathbf{z}) = \sum_{j=1}^{N} a_j \cdot 2(\mathbf{z} - \mathbf{v}_j) = \mathbf{0}$) yields $\mathbf{z} = \sum_{j=1}^{N} a_j \mathbf{v}_j$.

**Vulnerability analysis of vanilla attention.** When adversaries perturb the input tokens, these tokens will dominate the impact on output tokens since the quadratic penalty on the residual $\|\mathbf{v}_j - \mathbf{z}\|^2$ will dominate the WLS estimator. Therefore, the output token $\mathbf{z}$ will be shifted to those dominating input tokens. As a result, the adversarial input tokens will significantly impact the representation of output tokens. We also provide an empirical study to verify that adversarial attacks will significantly increase the residual $\|\mathbf{v}_j - \mathbf{z}\|^2$ in Appendix F.3. Moreover, we simulate a mean estimation problem under outlier data points using synthetic data to better illustrate the sensitivity of the WLS estimator. The detailed setting and visualization results of the numerical simulation are provided in Appendix F.

## 3.2 Robust WLS Token Estimators

The analysis above provides a valid explanation of why various attention-based transformer architectures are easily compromised by introducing adversarial perturbations in the input data. Also, our interpretation of the attention mechanism in transformers as WLS estimator provides a rigorous perspective to design robust alternatives. To dampen the effect of outlier data, multiple robust regression algorithms have been proposed in robust statistics using least absolute deviations [48], Huber regression [49], and Minimax Concave Penalty (MCP) [50]. Motivated by these advancements with rigorous robustness guarantees, we propose the robust weighted least squares token estimators to enhance the resilience against potential adversarial attacks as follows:

$$\arg\min_{\mathbf{z}} \mathcal{L}(\mathbf{z}) = \sum_{j=1}^{N} a_j \cdot \rho(\|\mathbf{v}_j - \mathbf{z}\|) \tag{2}$$

where $\rho$ can be flexibly replaced with the specific robust penalties in Figure 3.

**Special cases of $\rho$.** (1) The quadratic $\ell_2$ loss recovers vanilla WLS estimator; $\ell_1$ loss exerts linear effect on the residuals; (2) Huber loss performs as $\ell_2$ loss within the range $(0, \delta)$, and becomes similar to $\ell_1$ when $z > \delta$; (3) MCP loss behaves like $\ell_1$ loss near zero and becomes constant when $z$ is large than $\gamma$. (4) We also propose Huber-MCP to combine the advantage of Huber and MCP loss. The detailed formulations are available in Appendix B.4 due to the space limit.

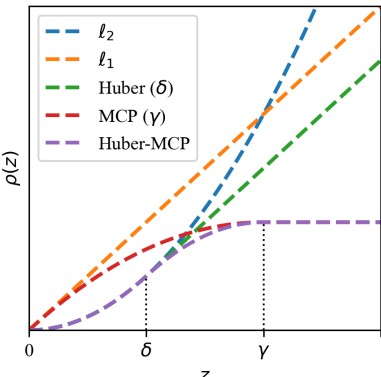

Figure 3: Different $\rho(z)$.

## 3.3 Newton-IRLS algorithm

The proposed robust token estimator in Eq. (2) is non-convex and non-smooth, posing a challenge for efficient algorithm design. Moreover, the exploding model size of evolving transformers further necessitates the design of efficient neural network layers. To this end, we propose an efficient Newton iterative reweighted least square (Newton-IRLS) algorithm to tackle this challenging problem. We first design a localized upper bound for the original objective and then optimize the upper bound with a second-order Newton method. We also provide a rigorous theoretical loss descent guarantee. The precise statements are presented as follows and the detailed proof are provided in Appendix B.

**Localized upper bound.** Instead of directly optimizing the original loss function $\mathcal{L}(\mathbf{z})$ in Eq. (2), we optimize a convex localized upper bound at the current iteration $\mathbf{z}^{(k)}$ as follows:

**Lemma 3.1** (Localized Upper Bound). *Suppose the loss objective is defined as in Eq. (2), where $\rho \circ sqrt(\cdot)$ is any non-convex function. For any fixed point $\mathbf{z}^{(k)}$, there exists a convex localized upper*

*bound as:*

$$\hat{\mathcal{L}}(\mathbf{z}) = \sum_{j=1}^{N} a_j \cdot w_j^{(k)} \cdot \|\mathbf{v}_j - \mathbf{z}\|^2 + C(\mathbf{z}^{(k)}), \tag{3}$$

*where $w_j^{(k)} = \frac{\rho'(\|\mathbf{v}_j - \mathbf{z}^{(k)}\|)}{2\|\mathbf{v}_j - \mathbf{z}^{(k)}\|}$ and $\rho'$ is the first derivative of $\rho$. Particularly, the constant $C(\mathbf{z}^{(k)})$ guarantees the equality of $\hat{\mathcal{L}}$ and $\mathcal{L}$ at $\mathbf{z}^{(k)}$, i.e., $\hat{\mathcal{L}}(\mathbf{z}^{(k)}) = \mathcal{L}(\mathbf{z}^{(k)})$.*

*Proof.* Please refer to Appendix B.1. □

As $C(\mathbf{z}^{(k)})$ is treated as a constant during the optimization at the current step, the upper bound in Eq. (3) becomes convex and can be efficiently optimized.

**Newton-IRLS iteration.** After obtaining the convex upper bound $\hat{\mathcal{L}}$ in Eq. (3), we can derive a concise closed-form iteration using the second-order Newton method as follows:

$$\mathbf{z}^{(k+1)} = \mathbf{z}^{(k)} - \left[\nabla^2 \hat{\mathcal{L}}(\mathbf{z}^{(k)})\right]^{-1} \nabla \hat{\mathcal{L}}(\mathbf{z}^{(k)}) = \frac{\sum_j a_j \cdot w_j^{(k)} \cdot \mathbf{v}_j}{\sum_j a_j \cdot w_j^{(k)}}. \tag{4}$$

Eq. (4) can be interpreted as a reweighted sum, in which the derived $w_j^{(k)}$ modifies the original attention score $a_j$ on the value vector $\mathbf{v}_j$. We leave detailed derivations of Newton-IRLS algorithm in Appendix B.2. Its convergence and rigorous loss descent are guaranteed by the following Theorem 3.2.

**Theorem 3.2** (Convergence guarantee)**.** *Suppose the loss objective $\mathcal{L}(\mathbf{z})$ is defined as in Eq. (2) and its corresponding convex localized upper bound is in Eq. (3). Then, through the iteration in Eq. (4), the following inequality holds:*

$$\mathcal{L}(\mathbf{z}^{(k+1)}) \leq \hat{\mathcal{L}}(\mathbf{z}^{(k+1)}) \leq \hat{\mathcal{L}}(\mathbf{z}^{(k)}) = \mathcal{L}(\mathbf{z}^{(k)}), \tag{5}$$

*that is, optimizing upper bound $\hat{\mathcal{L}}$ can guarantee the rigorous descent of $\mathcal{L}$.*

*Proof.* Please refer to Appendix B.3. □

Although the loss $\mathcal{L}(\mathbf{z})$ is not necessarily convex and does not possess a global optimum, Theorem 3.2 guarantees that the Newton-IRLS iteration, which optimizes $\hat{\mathcal{L}}(\mathbf{z})$, can rigorously reduce the original loss $\mathcal{L}(\mathbf{z})$. The algorithm analyses in Appendix F, along with the main experiments in Section 4 and Section 5, validate that the local optimal solution achieved by our algorithm performs well in terms of both convergence and empirical robustness.

**Robust token estimator by reweighting the tokens**. The robust estimator in Eq. (2) provides a general framework that covers several special cases. By choosing different penalty functions $\rho$ on the residuals $\|\mathbf{v}_j - \mathbf{z}^{(k)}\|$, we obtain various reweighting schemes in Eq. (4). Take the MCP function as the instance, the weight is derived as $w_j^{(k)} = \frac{\rho'_\gamma(\|\mathbf{v}_j - \mathbf{z}^{(k)}\|)}{2\|\mathbf{v}_j - \mathbf{z}^{(k)}\|} = \max\left[\frac{1}{\|\mathbf{v}_j - \mathbf{z}^{(k)}\|} - \frac{1}{\gamma}, 0\right]$. Obviously, the weight $w_j^{(k)}$ becomes smaller as $\|\mathbf{v}_j - \mathbf{z}^{(k)}\|$ increases, thereby down-weighting the large residuals. The residuals will be completely removed when it exceeds the threshold $\gamma$, since the weight then becomes 0. The complete discussions for all cases are provided in Appendix B.4.

### 3.4 ProAttention: Robust Attention Layers

In the previous subsection, we formulate the token-wise Newton-IRLS approach for notation simplicity. Here, we will present the corresponding matrix version for the entire attention layer.

**Matrix Form.** Denote $\mathbf{V} = \{\mathbf{v}_j\}_{j \in [N]}$ and $\mathbf{A} = \{a_{ij}\}_{i,j \in [N]}$ are value matrix and the attention matrix, respectively. $\mathbf{Z}^{(k)} = \{\mathbf{z}_i^{(k)}\}_{i \in [N]}$ is the estimator for token $i$ at the $k$-th iteration. Subsequently, the pairwise distance $\mathbf{D}^{(k)} = \{\|\mathbf{v}_j - \mathbf{z}_i^{(k)}\|\}_{i,j \in [N]}$ between $\mathbf{Z}^{(k)}$ and $\mathbf{V}$ can be efficiently computed using the `torch.cdist` function in PyTorch. Following this, the weight $\mathbf{W}^{(k)} = \{w_{ij}^{(k)}\}_{i,j \in [N]}$ can be calculated element-wise based on $\mathbf{D}^{(k)}$. Then the next step $\mathbf{Z}^{(k+1)}$ is updated as a reweighted matrix multiplication $(\mathbf{W}^{(k)} \odot \mathbf{A}) \cdot \mathbf{V}$.

**Plug-and-Play Robust Attention.** The proposed algorithm can be packaged as a robust attention module, which can be readily plugged into the transformers as a **P**lug-and-Play **Ro**bust Attention (**ProAttention**) layer without additional training or fine-tuning as shown in Figure 2. The implementation of ProAttention using MCP penalty in PyTorch is shown in Algorithm 1. The complete pseudocode for other penalties is presented in in Appendix A.

---
**Algorithm 1** ProAttention (MCP)

---
```
1 D = torch.cdist(Z, V) # Pairwise distance
2 W = torch.clip(1/D-1/gamma, min=0) # MCP
3 W = normalize(W * A, p=1, dim=-1)
4 Z = torch.matmul(W, V) # Update
```
---

**Complexity analysis.** Let $N$, $D$, and $K$ represent the length of tokens, the dimension of vectors, and the steps of the iterations, respectively. The vanilla attention requires $2 \cdot N \times N \times D$ basic operations while our ProAttention needs $(1 + 2K) \cdot N \times N \times D$. However, our ProAttention remains efficient, as the Newton-IRLS method can effectively approximate the solution within only 3 steps ($K \leq 3$) (Figure 4 (a)) and ProTransformers do not introduce additional computation for training or fine-tuning. We provide the detailed complexity analysis of various attentions in Appendix L.

**Advantages.** Our proposed ProAttention enjoys the following advantages: (1) *Simplicity*: it is simple and easy to implement with only 4 core lines of code in Algorithm 1; (2) *Efficiency*: it is a plug-and-play layer that can be integrated into any trained transformer without additional training or fine-tuning; (3) *Universality*: it is a universal framework that advances the vanilla attention mechanism into a series of robust derivatives with different penalties. Moreover, it can be applied to any attention-based model across various modalities and tasks.

In the following sections, we will present comprehensive experiments and studies to validate the effectiveness of the proposed ProAttention on language modeling in Section 4 as well as computer vision and graph learning in Section 5.

# 4   Experiment on Language Modeling

In this section, we evaluate the effectiveness of the proposed ProAttention and ProTransformer under classic text attacks on pre-trained language models, and two prompting-based attacks (prompt attack and jailbreak attack) in the context of LLMs with comprehensive ablation studies.

## 4.1   Experiment Setting

**Tasks and Datasets.** For topic classification, we use AG's News Corpus (AGNEWS) [51]. For sentiment analysis, we utilize two widely-used datasets: Internet Movie Database (IMDB) [52] and Stanford Sentiment Treebank (SST-2) [53]. For textual entailment, we make use of Recognizing Textual Entailment (RTE) in the General Language Understanding Evaluation benchmark [54]. For jailbreak attack, we select a new dataset Behaviors introduced in [55]. For the detailed information on these datasets, please refer to Appendix C.

**Backbone Architectures.** For classical pre-trained language models, we choose BERT [56] and its variants including RoBERTa [57], ALBERT [58] and DistilBERT [59]. For large language models (LLMs), we choose T5 [60], LLaMA [12] and Vicuna [13]. For the detailed information on backbone architectures, please refer to Appendix D.2.

**Attacks.** We not only evaluate several classic text attacks but also include popular prompt attacks and jailbreak attacks on the LLMs. The three attack mechanisms and their differences are illustrated in Figure 1. For classic text attacks, we evaluate the attacks at various levels, including the character-level DeepWordBug [7], word-level PWWS [9], TextFooler [14], and multi-level TextBugger [8]. For prompt attacks, we modify the prompt template according to the aforementioned text attacks following the evaluation setting in PromptBench [15]. For jailbreak, we evaluate the suffix attack using Greedy Coordinate Gradient (GCG) method [55] and we test both attacks transferred from surrogate model Vicuna (transfer attack) and attacks directly targeting the victim models (adaptive attack). Please refer to Appendix E for details on attacks.

**Defense Baselines**. We include the following defense baselines in our experiments: MixADA [17], PGD-Adv [31], FreeLB [47], TA-VAT [18] and SmoothLLM [61]. Additionally, we also include the adversarial training (AT), wherein the augmented perturbations are generated by the attack to be assessed. Details of these defense methods are provided in Appendix D.1.

**Evaluation metrics.** Following [62], we use 3 metrics to evaluate the model performance. Clean accuracy (**Clean%**) is the model accuracy on the clean testing data. Accuracy under attack (**AUA%**) is the accuracy on the perturbed data under specific attack. Attack success rate (**ASR%**) is the ratio of the number of successfully perturbed cases divided by the number of attempted texts.

**Hyperparameters.** For text attack setting, we follow the setting in the TextAttack framework [63]. For prompt attack, we follow the setting in PromptBench [15]. For GCG-based jailbreak attack, we follow the setting in [61]. The detailed attack settings can be found in Appendix E. For defense baselines, we follow the settings in their original papers. For our ProTransformer, we set the default number of ProAttention layers as $K = 3$ since it can quickly converge to a reasonable precision within 3 layers. Finally we tune $\delta$ (default 1) or $\gamma$ (default 4) in the penalties (Huber and MCP loss) to obtain the optimal parameters.

## 4.2 Classic Text Attacks on Language Models

To demonstrate the effectiveness of the proposed ProTransformer, we compare the robustness of our methods with several popular defenses in three classical tasks: topic classification, sentiment analysis, and textual entailment.

### 4.2.1 Adversarial Robustness

Table 1: The results of topic classification on AGNEWS.

| Model | Clean% ↑ | Textfooler Aua% ↑ | Textfooler ASR% ↓ | TextBugger AUA% ↑ | TextBugger ASR% ↓ | DeepWordBug AUA% ↑ | DeepWordBug ASR% ↓ | PWWS AUA% ↑ | PWWS ASR% ↓ |
|---|---|---|---|---|---|---|---|---|---|
| ALBERT | 93.0 | 20.6 | 77.9 | 26.1 | 71.9 | 38.9 | 58.2 | 35.9 | 61.4 |
| Pro-ALBERT (MCP) (Ours) | 93.8 | 48.9 | 47.3 | 41.8 | 55.3 | 59.5 | 35.9 | 63.1 | 32.0 |
| DistilBERT | 93.5 | 13.2 | 85.9 | 33.6 | 63.4 | 30.0 | 67.9 | 36.5 | 61.0 |
| Pro-DistilBERT (MCP) (Ours) | 93.9 | 29.3 | 68.5 | 48.7 | 47.9 | 34.3 | 63.1 | 50.5 | 45.6 |
| RoBERTa | 93.4 | 13.0 | 86.1 | 32.5 | 64.5 | 41.2 | 55.9 | 34.0 | 63.6 |
| Pro-RoBERTa (MCP) (Ours) | 93.7 | 24.4 | 73.7 | 34.3 | 62.8 | 45.5 | 51.5 | 39.4 | 57.5 |
| BERT | 94.2 | 19.7 | 78.9 | 31.7 | 67.5 | 37.5 | 59.8 | 43.1 | 53.8 |
| + FreeLB | 94.2 | 38.0 | 59.5 | 42.8 | 55.5 | 56.1 | 40.9 | 57.0 | 39.9 |
| + PGD | 94.1 | 36.8 | 61.7 | 40.5 | 57.1 | 47.6 | 49.7 | 48.7 | 48.6 |
| + MixADA | 94.3 | 35.6 | 62.4 | 35.4 | 62.9 | 38.2 | 50.5 | 46.8 | 50.4 |
| + TA-VAT | 94.4 | 36.2 | 61.8 | 39.2 | 58.2 | 49.5 | 48.1 | 47.0 | 50.7 |
| + AT | 94.1 | 42.1 | 54.8 | 56.1 | 39.4 | 42.4 | 54.1 | 62.6 | 32.5 |
| Pro-BERT ($\ell_1$) (Ours) | 94.2 | 23.8 | 74.5 | 43.8 | 53.0 | 48.7 | 47.8 | 46.5 | 50.1 |
| Pro-BERT (Huber) (Ours) | 94.2 | 24.2 | 74.0 | 43.7 | 52.9 | 46.0 | 50.5 | 48.4 | 47.9 |
| Pro-BERT (MCP) (Ours) | 93.2 | 39.2 | 57.7 | 48.3 | 48.5 | 51.8 | 43.8 | 56.2 | 39.2 |
| Pro-BERT (MCP) + AT (Ours) | 94.0 | 56.8 | 38.9 | 60.7 | 35.1 | 61.0 | 34.1 | 68.8 | 25.7 |

**Performance analysis.** The experimental results of topic classification (AGNEWS) are presented in Table 1, and we provide the results of sentiment analysis (IMDB) and textual entailment (RTE) in Appendix G.1 and G.2 due to the space limit. From the experiment results, we can make the following observations:

- The proposed ProAttention is a highly effective plug-in module that significantly and consistently enhances the robustness of various transformer backbones across various adversarial attacks. Taking AGNEWS as the instance, when combined with ProAttention (MCP), under the attacks {Textfooler, TextBugger, DeepWordBug, PWWS}: (1) ALBERT is improved by {28.3%, 15.7%, 20.6%, 27.2%} (2) DistilBERT is improved by {16.1%, 15.1%, 4.3%, 14.0%} (3) RoBERTa is improved by {11.4%, 1.8%, 4.3%, 5.4%} (4) BERT is improved by {19.5%, 16.6%, 14.3%, 13.1%}.

- Our method, Pro-BERT (MCP) + AT, exhibits best robustness among all the baselines. By simply plugging in ProAttention (MCP) module without fine-tuning, our Pro-BERT can achieve comparable robustness to most adversarial training-based methods which require substantial computational time and resources. Furthermore, our framework is orthogonal to most existing defenses, allowing for combined use with them to further enhance robustness. For instance, when combined with AT technique, our Pro-BERT (MCP) + AT can further improve BERT + AT by {14.7%, 4.6%, 18.6%, 6.2%} under {TextFooler, TextBugger, DeepWordBug, PWWS}.

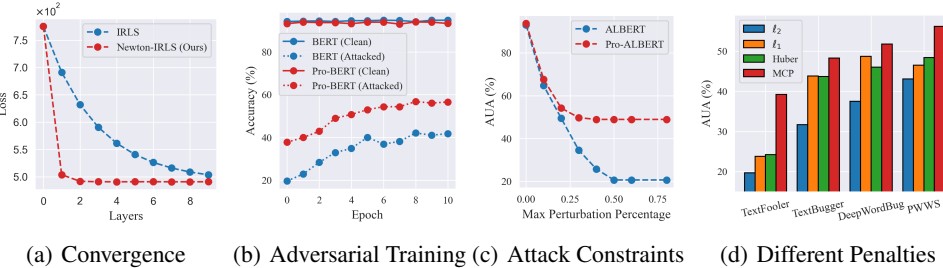

| (a) Convergence | (b) Adversarial Training | (c) Attack Constraints | (d) Different Penalties |

Figure 4: Ablation studies.

#### 4.2.2 Ablation Study

**Convergence.** To validate the advantage of our Newton-IRLS over the first-order method, we conduct a simulation experiment and plot the loss descent curves in Figure 4 (a). It can be observed that Newton-IRLS exhibits efficient convergence as claimed in Section 3.3. We provide the experiment details, loss descent curves (Figure 7), and the visualization of trajectories (Figure 8) of the updated vectors in 2D plane in Appendix F.1 to further demonstrate the effectiveness of our algorithm.

**Adversarial fine-tuning.** To get insight into how the models gain more robustness from adversarial examples, we track the training curves of adversarial fine-tuning under TextFooler in Figure 4 (b), and put the results of other attacks in Figure 10 in Appendix G.3. We can observe that our Pro-BERT (MCP) is compatible with adversarial fine-tuning technique to further enhance the model resilience.

**Attack constraints.** In text attack, there are several kinds of attack constraints including the maximum percentage of perturbed words, minimum cosine similarity between the replaced synonym and original word, and minimum sentence similarity threshold between the original sentence and perturbed sentence. We test the values of these constraints in TextFooler. We present the results under different perturbation percentages in Figure 4 (c) and other constraint measurements in Appendix G.4. From the results, we observe that our Pro-ALBERT (MCP) can significantly outperform the backbone ALBERT across all ranges of constraints.

**Different penalties.** Our Newton-IRLS is flexible to be formulated as different robust estimators with different penalties. From the comparison in Figure 4 (d) , it can be observed that our robust framework can consistently improve the robustness of the backbone BERT ($\ell_2$). Specifically, $\ell_1$ and Huber-based defenses are comparable, and MCP-based method exhibits the best performance.

**Different backbones.** Our method is a general plug-and-play layer applicable to various transformer backbones. The results in Table 1 and the ablation study on different backbones in Appendix G.5 (Figure 12) demonstrate that ProAttention improves the robustness over various architecture backbones (BERT, RoBERTa, DistilBERT and ALBERT) against various attacks with significant margins.

**Running time.** To empirically evaluate the efficiency of our method, we test the average running time on AGNEWS using BERT and Pro-BERT (MCP) equipped with multi-layer ($K$) ProAttention. The results in Table 2 show that our ProAttention only requires 1-2 times additional inference time of the backbone model yet achieves significant improvement in robustness without training.

Table 2: Average running time (ms) on AGNEWS.

|  | BERT | Pro-BERT (MCP) | | | | | |
|---|---|---|---|---|---|---|---|
| Running time (ms) | 6.14 | 9.04 | 11.67 | 14.34 | 17.33 | 19.89 | 21.87 |
| # Layers ($K$) | \ | 1 | 2 | 3 | 4 | 5 | 6 |

### 4.3 Adversarial Prompting Attacks on LLMs

In the context of prompt-based generative AI, the adversarial attacks mechanisms on LLMs become more enriched and sophisticated. In this section, we will evaluate the robustness of our proposed ProTransformer under two popular attacks: *prompt attack* and *jailbreak attack*.

### 4.3.1 Prompt Attack

As shown in Figure 1, the most significant distinction between prompt attacks and classical text attacks is that prompt attacks aim to mislead the models by altering the prompt template rather than the input content. We display the results of T5 in Figure 5 and leave the comprehensive study in Appendix H.1. We also present the results on LLaMA in Appendix H.2. From the results, we can make the following observations: (1) For T5, the choice of the penalty would affect the robustness of defenses. Specifically, Pro-T5 (MCP) exhibits a significant advantage over other methods, and this advantage becomes even more evident as the number of perturbed words increases. Pro-T5 ($\ell_1$) and Pro-T5 (Huber) show a slight improvement over the backbone model T5. (2) For LLaMA, Huber-MCP and Huber-based methods exhibit better robustness than other methods while preserving good clean performance. The detailed experiments and discussions can be found in Appendix H.2.

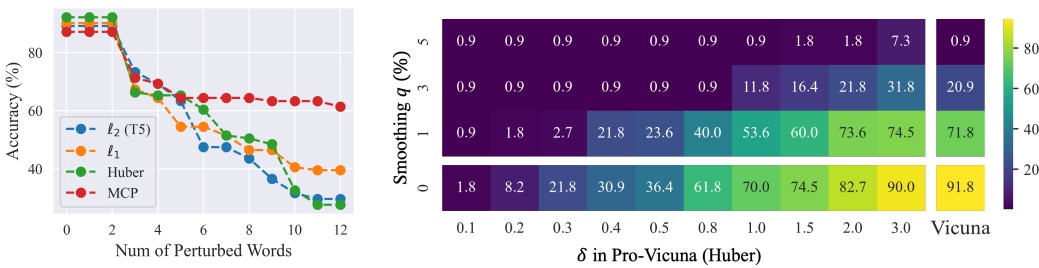

Figure 5: Prompt attack results. Figure 6: Attack success rates (ASRs) under transfer jailbreak.

### 4.3.2 Jailbreak Attack

In recent years, prompts have played a pivotal role in guiding models to generate desired outputs. Nevertheless, there exist malicious "jailbreak prompts", which are intentionally designed to bypass the built-in safeguards in LLMs, causing the model to produce harmful content that violates the legal policies. As illustrated in Figure 1, the suffix-injection jailbreaks attempt to append a non-semantic suffix to the user's prompt to fool the models. We select GCG method to evaluate the resilience of models comprehensively.

In Figure 6, we compare the Attack Success Rates (ASRs) of Vicuna and its corresponding Pro-Vicuna (Huber) with various $\delta$ values on Behaviors. In each column, we also include SmoothLLM [61] with different smoothing extent $q(\%)$ to further reinforce the resilience of every single model. The last row of matrix ($q = 0$) stands for the performance without random smoothing. The additional results of random smoothing with swap, insert and patch, as well as the results under adaptive jailbreaking attack are presented in Appendix I.

From the results, we can observe that: (1) Our Pro-Vicuna can significantly improve the robustness of Vicuna. As shown in the last row of Figure 6, with $\delta = 0.1$, we successfully reduce the ASR to 1.8%, which is comparable to the random smoothing defense that requires multiple random perturbations, inferences and aggregations. (2) Our ProAttention is orthogonal to randomized smoothing defense and can be combined with it to further improve the robustness.

## 5 Experiment beyond Language Modeling

In the previous section, we have provided comprehensive experiments to validate the effectiveness of our ProTransformer in the (large) language models. In fact, as shown in Figure 2, our ProAttention is a fundamental module which can reinforce any attention-based models across various domains or modalities. In this section, we will integrate ProAttention into vision models and graph learning models to further validate the effectiveness and generality of our approach.

## 5.1 Image Classification

In computer vision, we conduct two attacks (FGSM [30] and PGD [31]) on several vision transformers including ViT, BeiT, ConviT, DeiT and Swin. We perform the experiments on CIFAR-10 and ImageNet-1K across budgets $\{1/255, 4/255, 8/255\}$, and present the results of PGD on CIFAR-10 in Table 3 and additional experiments in Appendix J. From the results, we can observe that Pro-ViT can outperform the second best model by $\{35.64\%, 46.28\%, 33.14\%\}$ under different budgets.

Table 3: Adversarial robustness under PGD.

| Model\Budget | 0 (Clean) | 1/255 | 4/255 | 8/255 |
|---|---|---|---|---|
| Deit | 97.91 | 38.98 | 0.44 | 0.0 |
| Convit | 98.70 | 41.75 | 1.83 | 0.0 |
| BeiT | 97.87 | 6.81 | 0.0 | 0.0 |
| Swin | 98.30 | 14.89 | 0.02 | 0.01 |
| ViT | **98.74** | 34.61 | 1.83 | 0.26 |
| Pro-ViT (Ours) | 98.40 | **77.39** | **48.11** | **33.40** |

## 5.2 Graph Representation Learning

Besides the language and vision domains, we also validate the effectiveness of our method in the graph domain. We conduct the semi-supervised node classification task and leverage PGD adaptive attack [64] to evaluate the robustness of models. We show the experiment results of Cora-ML and Citeseer, averaged over 5 different random splits, in Table 4 and Table 29 (in Appendix K), respectively. The ablation studies on the layers and $\gamma$ in MCP are presented in Table 30. Please refer to Appendix K for more

Table 4: Adversarial robustness on Cora-ML.

| Model \ Budget | 0% | 10% | 20% | 30% | 40% |
|---|---|---|---|---|---|
| GCN | 85.0 ± 0.4 | 69.6 ± 0.5 | 60.9 ± 0.7 | 54.2 ± 0.6 | 48.4 ± 0.5 |
| GNNGuard | 83.1 ± 0.7 | 70.2 ± 1.0 | 63.1 ± 1.1 | 57.5 ± 1.6 | 51.0 ± 1.2 |
| RGCN | 85.7 ± 0.4 | 69.1 ± 0.4 | 59.8 ± 0.7 | 52.8 ± 0.7 | 46.1 ± 0.7 |
| GRAND | **86.1 ± 0.7** | 70.7 ± 0.7 | 61.6 ± 0.7 | 56.7 ± 0.8 | 51.9 ± 0.9 |
| ProGNN | 85.6 ± 0.5 | 71.0 ± 0.5 | 63.0 ± 0.7 | 56.8 ± 0.7 | 51.3 ± 0.6 |
| Jaccard-GCN | 83.7 ± 0.7 | 68.3 ± 0.7 | 60.0 ± 1.1 | 54.0 ± 1.7 | 49.1 ± 2.4 |
| SoftMedian | 85.0 ± 0.7 | 75.5 ± 0.9 | 69.5 ± 0.5 | 62.8 ± 0.8 | 58.1 ± 0.7 |
| GAT | 83.5 ± 0.5 | 71.2 ± 1.2 | 65.0 ± 0.9 | 60.5 ± 0.9 | 56.7 ± 0.9 |
| Pro-GAT (ours) | 84.6 ± 0.8 | **75.5 ± 0.8** | **72.1 ± 0.4** | **69.0 ± 0.7** | **66.5 ± 1.2** |

detailed results and studies. From the results, we can conclude that our Pro-GAT significantly outperforms the backbone GAT and exhibits strong robustness across various budgets while keeping good clean accuracy.

## 6 Conclusion & Limitation

In this paper, we delve into the robustness and security of the popular transformer-based architectures. We revisit the vulnerability of attention mechanisms with theoretical understanding and simulations. We propose an interpretable robust attention layer to robustify transformer architecture via a plug-and-play paradigm. Our proposed ProAttention is an effective, efficient, and universal framework that can significantly enhance the robustness of transformers across various tasks, architectures, attacks, and domains without additional training or fine-tuning.

Regarding the limitations, despite the acceptable complexity of our ProTransformer, there is still potential to improve the efficiency of our models. Additionally, while we primarily claim and validate the effectiveness of our model under a plug-and-play paradigm, we are excited about the future of the proposed ProTransformer architecture and hope to see its full potential realized through training or fine-tuning on large models in the future.

## Acknowledgment

Zhichao Hou, Weizhi Gao, and Dr. Xiaorui Liu are supported by the National Science Foundation (NSF) National AI Research Resource Pilot Award, Amazon Research Award, NCSU Data Science Academy Seed Grant Award, and NCSU Faculty Research and Professional Development Award.

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

# A Pseudocode of Plug-and-Play Robust Attention (ProAttention)

Here, we provide the complete pseudocode of our ProAttention including various penalties cases in Algorithm 2. The core itertions are show in the for loop in the code. Our ProAttention is easy to implement by only replacing the vanilla attention module with our ProAttention.

**Algorithm 2** ProAttention in PyTorch style

```
1 class ProAttention(nn.Module):
2     def __init__(self, K, gamma, delta, penalty):
3         super().__init__()
4         self.K = K
5         self.gamma = gamma
6         self.delta = delta
7         self.penalty = penalty
8
9     def forward(self, A, V):
10        Z = torch.matmul(A, V) # Initialization
11
12        if self.penalty == 'L2':
13            return Z # Original attention
14
15        for _ in range(self.K):
16            D = torch.cdist(Z, V) # Pairwise distance
17
18            if self.penalty == 'L1':
19                W = 1/D
20            elif self.penalty == 'MCP':
21                W = torch.clip(1/D - 1/self.gamma, min=0)
22            elif self.penalty == 'Huber':
23                W = torch.clip(self.delta/D, max=1)
24            elif self.penalty == 'Huber-MCP':
25                W = torch.clip(self.delta/(self.gamma-self.delta)*(self.gamma/D-1), min=0, max=1)
26
27            W = torch.nn.functional.normalize(W * A, p=1, dim=-1) # Normalization
28            Z = torch.matmul(W, V) # Update
29        return Z
```

# B Proof of Newton-IRLS Algorithm

## B.1 Proof of Localized Upper Bound (Lemma 3.1)

*Proof.* Define $\phi(z) := \rho(\sqrt{z})$ as a non-convex function, then for fixed point $z_0$,

$$\phi(z) \leq \phi(z_0) + \phi'(z_0)(z - z_0) = \phi'(z_0) \cdot z + C(z_0)$$

where the first inequality holds with equality at $z = z_0$ and

$$\phi'(z_0) = \rho'(\sqrt{z}) \cdot \frac{1}{2\sqrt{z}}\bigg|_{z=z_0} = \frac{\rho'(\sqrt{z_0})}{2\sqrt{z_0}}.$$

By replacemnet as $z = \|\mathbf{v}_j - \mathbf{z}\|^2$ and $z_0 = \|\mathbf{v}_j - \mathbf{z}^{(k)}\|^2$, then

$$\rho(\|\mathbf{v}_j - \mathbf{z}\|) \leq \frac{\rho'(\|\mathbf{v}_j - \mathbf{z}^{(k)}\|)}{2\|\mathbf{v}_j - \mathbf{z}^{(k)}\|} \cdot \|\mathbf{v}_j - \mathbf{z}\|^2 + C(\|\mathbf{v}_j - \mathbf{z}^{(k)}\|^2)$$
$$= w_j^{(k)} \cdot \|\mathbf{v}_j - \mathbf{z}\|^2 + C(\|\mathbf{v}_j - \mathbf{z}^{(k)}\|^2),$$

and the first inequality holds with equality at $\mathbf{z} = \mathbf{z}^{(k)}$. Sum up the items on both sides with weights $\{a_j\}_{j \in [N]}$, we obtain

$$
\begin{aligned}
\mathcal{L}(\mathbf{z}) &= \sum_{j=1}^{N} a_i \cdot \rho(\|\mathbf{v}_j - \mathbf{z}\|) \\
&\leq \sum_{j=1}^{N} a_j \cdot w_j^{(k)} \cdot \|\mathbf{v}_j - \mathbf{z}\|^2 + \sum_{j=1}^{N} a_j \cdot C(\|\mathbf{v}_j - \mathbf{z}^{(k)}\|^2) \\
&= \sum_{j=1}^{N} a_j \cdot w_j^{(k)} \cdot \|\mathbf{v}_j - \mathbf{z}\|^2 + C_1(\mathbf{z}^{(k)}) \\
&= \hat{\mathcal{L}}(\mathbf{z})
\end{aligned}
\tag{6}
$$

and the equality holds at $\mathbf{z} = \mathbf{z}^{(k)}$:

$$
\hat{\mathcal{L}}(\mathbf{z}^{(k)}) = \mathcal{L}(\mathbf{z}^{(k)}).
\tag{7}
$$

$\square$

After obtaining the convex upper bound $\hat{\mathcal{L}}(\mathbf{z})$, it becomes feasible to employ convex optimization algorithms to optimize this objective.

## B.2 Proof of Newton-IRLS algorithm and Special Cases

**Newton-IRLS.** We first derive the formulations of gradient and Hessain matirx of $\hat{\mathcal{L}}$ as follows:

$$
\nabla \hat{\mathcal{L}}(\mathbf{z}^{(k)}) = \sum_{j=1}^{N} a_j \cdot w_j^{(k)} \cdot 2(\mathbf{z}^{(k)} - \mathbf{v}_i)
$$

$$
\nabla^2 \hat{\mathcal{L}}(\mathbf{z}^{(k)}) = \sum_{j=1}^{N} a_j \cdot w_j^{(k)} \cdot 2 \cdot \mathbf{I}
$$

Then, the gradient descent (GD) ($\eta$ is the stepsize) is

$$
\begin{aligned}
\mathbf{z}^{(k+1)} &= \mathbf{z}^{(k)} - \eta \cdot \nabla \hat{\mathcal{L}}(\mathbf{z}^{(k)}) \\
&= \mathbf{z}^{(k)} - \eta \cdot \sum_{j=1}^{N} a_j \cdot w_j^{(k)} \cdot 2(\mathbf{z}^{(k)} - \mathbf{v}_j),
\end{aligned}
$$

and the Newton Iteration is

$$
\mathbf{z}^{(k+1)} = \mathbf{z}^{(k)} - \left[ \nabla^2 \hat{\mathcal{L}}(\mathbf{z}^{(k)}) \right]^{-1} \nabla \hat{\mathcal{L}}(\mathbf{z}^{(k)})
\tag{8}
$$

$$
= \mathbf{z}^{(k)} - \cdot \left( \sum_{j=1}^{N} a_j \cdot w_j^{(k)} \cdot 2 \cdot \mathbf{I} \right)^{-1} \sum_{j=1}^{N} a_j \cdot w_j^{(k)} \cdot 2(\mathbf{z}^{(k)} - \mathbf{v}_i)
\tag{9}
$$

$$
= \frac{\sum_j a_j \cdot w_j^{(k)} \cdot \mathbf{v}_j}{\sum_j a_j \cdot w_j^{(k)}}
\tag{10}
$$

In convex optimization, it has been well-established that second-order methods converge much faster than first-order approaches, but they require substantial computation in calculating or approximating the inverse Hessian matrix. However, due to the uniqueness of our $\hat{\mathcal{L}}$ in Eq. (3), we can derive a concise closed-form iteration using the second-order Newton method as in Eq. (10). Compared to the first-order gradient descent (GD) iteration, our Newton-IRLS algorithm enjoys several advantages as follows:

- Fast convergence: Newton method converges at a quadratic rate, which is significantly faster than the linear convergence of gradient descent (GD). The comparative analysis of them can be found in Figure 4 (a) in ablation studies;

- Interpretable formulation: The resulted form in Eq. (**??**) employs a normalized reweighted sum, which can be interpreted as robust estimator by down-weighting the outliers, as discussed in the following paragraph;

- Efficient computation: The Hessian $\nabla^2 \hat{\mathcal{L}}(\mathbf{z}^{(k)})$ can be easily computed as a closed-form diagonal matrix, facilitating the matrix inversion and multiplication in the Newton's iteration.

## B.3 Proof of Rigorous Loss Descent Guarantee (Theorem 3.2)

*Proof.* Since $\mathbf{z}^{(k+1)}$ is obtained from optimize the convex localized upper bound $\mathcal{L}$ at $\mathbf{z}^{(k)}$, then we have $\hat{\mathcal{L}}(\mathbf{z}^{(k+1)}) \leq \hat{\mathcal{L}}(\mathbf{z}^{(k)})$. According to the upper bound in Eq. (6) and localized equality in Eq. (7), it is not hard to get the following inequality:

$$\mathcal{L}(\mathbf{z}^{(k+1)}) \leq \hat{\mathcal{L}}(\mathbf{z}^{(k+1)}) \leq \hat{\mathcal{L}}(\mathbf{z}^{(k)}) = \mathcal{L}(\mathbf{z}^{(k)}).$$

Therefore, optimizing the localized upper bound $\hat{\mathcal{L}}$ can guarantee the rigorous descent of $\mathcal{L}$.

$\square$

## B.4 Special cases of Newton-IRLS

Our Newton-IRLS is a general framework which can be derived as different reweighting schemes with different penalties:

- Square Loss ($\ell_2$):

$$\rho(z) = \frac{1}{2}z^2,$$

$$w_j^{(k)} = \frac{\rho'(\|\mathbf{v}_j - \mathbf{z}^{(k)}\|)}{2\|\mathbf{v}_j - \mathbf{z}^{(k)}\|} = \frac{1}{2},$$

$$\mathbf{z}^* = \sum_{j=1}^{N} a_j \cdot \mathbf{v}_j.$$

$\ell_2$ loss increase quadratically with $z$, which suggests that $\ell_2$ loss is more sensitive to the residual magnitude. Particularly, $\ell_2$ loss can recover the vanilla attention since the weights are constant $\frac{1}{2}$.

- Absolute Loss ($\ell_1$):

$$\rho(z) = z,$$

$$w_j^{(k)} = \frac{\rho'(\|\mathbf{v}_j - \mathbf{z}^{(k)}\|)}{2\|\mathbf{v}_j - \mathbf{z}^{(k)}\|} = \frac{1}{2\|\mathbf{v}_j - \mathbf{z}^{(k)}\|}.$$

With $\ell_1$ loss, the weight is inversely proportional to $\|\mathbf{v}_j - \mathbf{z}^{(k)}\|$. By up-weighting the inliers and down-weighting the outliers, $\ell_1$-based estimators can mitigate the effect of large magnitude residues.

- Minimax Concave Penalty (MCP) [50]:

$$\rho_\gamma(z) = \begin{cases} z - \frac{z^2}{2\gamma} & \text{if } y < \gamma \\ \frac{\gamma}{2} & \text{if } y \geq \gamma \end{cases},$$

$$w_j^{(k)} = \frac{\rho'(\|\mathbf{v}_j - \mathbf{z}^{(k)}\|)}{2\|\mathbf{v}_j - \mathbf{z}^{(k)}\|} = \frac{1}{2} \max\left[\frac{1}{\|\mathbf{v}_j - \mathbf{z}^{(k)}\|} - \frac{1}{\gamma}, 0\right].$$

MCP loss becomes constant when $z$ is large and the weight derived by MCP loss enhances the interpretability of the robust estimator by down-weighting or completely removing the outliers. To be specific, the weight $w_j$ becomes smaller as the distance $\|\mathbf{v}_j - \mathbf{z}^{(k)}\|$ increases, thereby down-weighting the outlying cases. When this distance exceeds the threshold $\gamma$, the weight becomes 0, totally removing the outliers.

- Huber loss:
$$\rho_\delta(z) = \begin{cases} \frac{1}{2}z^2 & \text{if } z < \delta \\ \delta \cdot (z - \frac{1}{2}\delta) & \text{if } z \geq \delta \end{cases},$$

$$w_j^{(k)} = \frac{\rho'(\|\mathbf{v}_j - \mathbf{z}^{(k)}\|)}{2\|\mathbf{v}_j - \mathbf{z}^{(k)}\|} = \frac{1}{2}\min\left[1, \frac{\delta}{\|\mathbf{v}_j - \mathbf{z}^{(k)}\|}\right].$$

Huber loss is equivalent to the $\ell_2$ loss within the range $(0, \delta)$, and it becomes similar to $\ell_1$ when $z > \delta$, which indicates that Huber loss may mitigate the effect of large noise while keeping decent performance in noiseless scenario.

- Huber-MCP:
$$\rho_{\delta,\gamma}(z) = \begin{cases} \frac{1}{2}z^2 & \text{if } z < \delta \\ \delta \cdot (z - \frac{1}{2}\delta - \frac{(z-\delta)^2}{2(\gamma-\delta)}) & \text{if } \delta \leq z < \gamma \\ \frac{\delta\gamma}{2} & \text{if } \gamma \leq z \end{cases},$$

$$w_j^{(k)} = \frac{\rho'(\|\mathbf{v}_j - \mathbf{z}^{(k)}\|)}{2\|\mathbf{v}_j - \mathbf{z}^{(k)}\|} = \frac{1}{2}\max\left[\min\left[\frac{\delta}{\gamma-\delta}\left(\frac{\gamma}{\|\mathbf{v}_j - \mathbf{z}^{(k)}\|} - 1\right), 1\right], 0\right].$$

This penalty combines the advantages of Huber and MCP in recovering the $\ell_2$ loss and largely mitigating the outliers.

# C  Dataset Information

## C.1  Language Domain

- **AG's News Corpus (AGNEWS)** [51]: It is a collection of more than 1 million news articles. News articles have been gathered from more than 2000 news sources by ComeToMyHead in more than 1 year of activity. ComeToMyHead is an academic news search engine which has been running since July, 2004. The dataset is provided by the academic comunity for research purposes in data mining (clustering, classification, etc), information retrieval (ranking, search, etc), xml, data compression, data streaming, and any other non-commercial activity. The AG's news topic classification dataset is constructed by choosing 4 largest classes from the original corpus. Each class contains 30,000 training samples and 1,900 testing samples. The total number of training samples is 120,000 and testing 7,600.

- **Internet Movie Database (IMDB)** [52]: IMDB dataset having 50K movie reviews for natural language processing or Text analytics. This is a dataset for binary sentiment classification containing substantially more data than previous benchmark datasets. We provide a set of 25,000 highly polar movie reviews for training and 25,000 for testing. So, predict the number of positive and negative reviews using either classification or deep learning algorithms.

- **Stanford Sentiment Treebank (SST-2)** [53]: It is a corpus with fully labeled parse trees that allows for a complete analysis of the compositional effects of sentiment in language. The corpus consists of 11,855 single sentences extracted from movie reviews. It was parsed with the Stanford parser and includes a total of 215,154 unique phrases from those parse trees, each annotated by 3 human judges. Binary classification experiments on full sentences (negative or somewhat negative vs somewhat positive or positive with neutral sentences discarded) refer to the dataset as SST-2 or SST binary.

- **Recognizing Textual Entailment (RTE)**: It comes from a series of annual textual entailment challenges. The authors of the benchmark combined the data from RTE1 [65], RTE2 [66], RTE3 [67], and RTE5 [68]. Examples are constructed based on news and Wikipedia text. The authors of the benchmark convert all datasets to a two-class split, where for three-class datasets they collapse neutral and contradiction into not entailment, for consistency.

- **Behaviors**: It is a new dataset introduced in [55] for robustness evaluation of jailbreaking attack. The dataset includes 520 goal prompts and corresponding targets, it is available in https://github.com/llm-attacks/llm-attacks/blob/main/data/advbench/.

## C.2  Beyond Language Domain

- **CIFAR10** [69]: The CIFAR-10 dataset is a well-known dataset used in the field of computer vision. It consists of 60,000 32x32 color images in 10 different classes, with 6,000 images per class. The dataset is divided into two parts: 50,000 training images and 10,000 test images. The 10 different classes represent airplanes, cars, birds, cats, deer, dogs, frogs, horses, ships, and trucks. Each image is labeled with one of these 10 categories.

- **ImageNet-1K** [70]: This dataset provides access to ImageNet which is the most commonly used subset of ImageNet. This dataset spans 1000 object classes and contains 1,281,167 training images, 50,000 validation images and 100,000 test images. The version also has the patch which fixes some of the corrupted test set images already applied.

- **Cora-ML** [71]: The Cora dataset is a widely-used benchmark dataset in the field of graph-based tasks. It consists of 2708 scientific publications classified into one of seven classes. The citation network consists of 5429 links. Each publication in the dataset is described by a 0/1-valued word vector indicating the absence/presence of the corresponding word from the dictionary. The dictionary consists of 1433 unique words. Working with the Cora dataset presents challenges typical of real-world graph data, such as handling sparse and high-dimensional feature vectors, and dealing with the complex structure of the graph.

- **Citeseer** [72]: The CiteSeer dataset is another popular dataset in the graph field. It consists of 3312 scientific publications classified into one of six classes. The citation network consists of 4732 links. Each publication in the dataset is described by a 0/1-valued word vector indicating the absence/presence of the corresponding word from the dictionary. The dictionary consists of 3703 unique words.

# D   Defense Baselines and Backbone Architectures

## D.1   Defense Baselines

**Language Domain:**

- **PGD-Adv** [31]: The Projected Gradient Descent (PGD) method stands as the most prevalent attack strategy in the field of computer vision. It is primarily utilized for crafting adversarial examples in the context of adversarial training. The defense in this paper is adapted directly from PGD-adv in computer vision, extending its application to language modeling.

- **MixADA** [17]: The search space for adversarial examples in language models is typically vast due to their discrete nature. To enhance the robustness of these models, MixADA integrates adversarial training [30] with mixup data augmentation [73], thereby expanding the range of adversarial examples covered. Specifically, mixup generates synthetic training examples by linearly blending pairs of inputs and their corresponding labels. This approach enables the model to learn from a broader and more effective set of adversarial examples during training.

- **FreeLB** [47]: Different from attacks that directly change the words in the sentence, FreeLB adds adversarial perturbations to word embeddings and minimizes the resultant adversarial loss around input samples. To expedite the process of adversarial training, FreeLB implements a single descent step on the parameters concurrently with each of the $K$ ascent steps applied to the perturbation, which utilizes the average of accumulated gradients over the $K$ steps. This efficiency has established FreeLB as a popular defense method in the field of NLP.

- **TA-VAT** [18]: TA-VAT is another virtual adversarial training method that generates gradient-based perturbations on the embedding space. To create fine-grained perturbations, TA-VAT employs a token-level accumulated perturbation vocabulary. This vocabulary serves to better initialize the perturbations. Additionally, TA-VAT utilizes a token-level normalization ball, which effectively constrains these perturbations in a relevant and precise manner.

- **Adversarial Training (AT)**: Adversarial training is adaptive to the attack to be evaluated. Take the Textfooler as the instance, at every epoch, we generate 1000 perturbations from the Textfooler and add them into the training dataset to reinforce the training of models. We utilize the TextAttack [63] platform the conduct this adversarial training.

- **SmoothLLM** [61]: Motivated by finding that the adversarial-prompting jailbreak is sensitive to the random character-level changes, SmoothLLM is designed by firstly perturbing multiple copies of the given prompt and then aggregating all the outputs.

**Beyond Language Domain:**

- **Graph Convolutional Network (GCN)** [74]: GCN is motivated by the localized first-order approximation of spectral graph convolutions. The basic idea is to first add self-loops to the adjacency matrix and then normalize the matrix.

- **Graph Attention Network (GAT)** [5]: GAT leverages the attention mechanism to construct masked self-attentional layers. This allows the nodes to reweight their neighbors via the feature similarity.

- **GNNGuard** [75]: GNNGuard is a universal reweighting framework that can be applied to any GNN. It leverages the cosine similarities between nodes' features to up-weight the correlated nodes and prune the edges between the dissimilar pairs.

- **Robust GCN (RGCN)** [76]: RGCN first models the latent representations as the Gaussian distributions. Then the weights of different neighborhoods will be assigned different weights according to their variances when performing the message propagation.

- **Graph Random Neural Network (GRAND)** [77]: The core of GRAND is the random propagation, wherein the node feature will be partially or entirely dropped out and then propagated through over the graph. This operation enable the node to be insensitive to the specific neighborhood, which prevents the effect of malicious outliers. Additionally, the random propagation also help to augment the representation for each node, thus improving the generalization of GNN.

- **Property GNN (ProGNN)** [78]: The core principle of ProGNN is to robustify the GNNs through enhancing the graph properties of sparsity, low rank and feature smoothness. It provides a graph structure learning framework to learn the clean graph structure and parameters simultaneously.

- **Jaccard-GCN** [79]: The basic idea of Jaccard-GCN is to preprocess the adjacency matrix by first computing the Jaccard coefficients of paired node features and then dropping the edges where the coefficients are below the threshold.

- **SoftMedian** [80]: SoftMedian is a robust estimator for the message passing aggregation. It reweights the adjacency weights based to the distances of the hidden embeddings between the neighbor nodes and the dimension-wise median of the the entire neighboring representations.

## D.2 Backbone Architectures

**Classical language models:**

- **BERT** [56]: BERT stands out as one of the most well-known transformer-based language models. It is pretrained through masked language modeling (MLM), where it learns to predict words that have been masked, using context for guidance. This pretrained model is then fine-tuned for a variety of downstream tasks, showcasing its versatility and effectiveness in diverse applications. In our experiments, we will use BERT-110M.

- **RoBERTa** [57]: RoBERTa is developed to overcome certain limitations of the original BERT model. This is accomplished by implementing key modifications such as increasing the batch size, extending the training epochs, and employing advanced optimization techniques. As a result of these strategic changes, RoBERTa has demonstrated substantial performance improvements over BERT across various NLP benchmarks. In our experiments, we will use RoBERTa-125M.

- **ALBERT** [58]: ALBERT is a lite variant of BERT. It is achieved by decoupling the word embedding from the hidden embedding, significantly cutting down the number of parameters. To further enhance its efficiency, ALBERT employs cross-layer parameter sharing, ensuring that all layers use the same parameters. The reductions not only minimize memory footprint but also improve the efficiency of the model. In our experiments, we will use ALBERT-12M.

- **DistilBERT** [59]: DistilBERT is a light version of BERT, maintaining most of the performance of the original BERT. It is trained with the knowledge distillation technique [81] to achieve high efficiency. In our experiments, we will use DistilBERT-66M.

**Large Language Models:**

- **T5** [60]: Text-to-Text Transfer Transformer (T5) is a transformer-based neural network model known for its versatility and power in handling a wide range of NLP tasks. T5 simplifies NLP tasks by treating them uniformly as text-generation challenges. The T5 model family offers a range of sizes, from 60 million to 11 billion parameters, catering to different computational needs. The flexibility has made T5 a popular choice in NLP research. In our experiments, we will use T5-770M.

- **LLaMA** [12]: LLaMa, the Large Language Model developed by Meta AI, represents a cutting-edge advancement in language modeling. Trained on publicly available datasets, LLaMa is available in various sizes to suit different computational needs. Notably, LLaMa-13B demonstrates superior performance over GPT-3 in most benchmarks, highlighting its exceptional effectiveness and capability in NLP tasks. In our experiments, we will use LLaMA-7B.

- **Vicuna** [13]: Vicuna is a high-performing, open-source chatbot that impresses with capabilities comparable to GPT-4. Fine-tuned from the LLaMa model, it utilizes user-shared conversations gathered from Share-GPT for its training. Remarkably, Vicuna achieves 90% of the performance level of GPT-4, despite having only 13 billion parameters, showcasing its efficiency and effectiveness. In our experiments, we will use Vicuna-7B.

**Vision Models:**

- **ViT** [3]: The Vision Transformer (ViT) is a model in computer vision that adopts the principles of the Transformer architecture. In ViT, an image is processed similarly to a sequence of words, or tokens. Specifically, the image is segmented into fixed-size patches, each of which is then linearly transformed into an embedded representation. When trained on sufficient data, ViT achieves state-of-the-art performance on image classification benchmarks, competing with or outperforming leading CNN-based models. In our experiments, we will use ViT-86M.

- **Swin** [4]: Swin Transformer is a popular variant of ViT, standing out for its enhanced efficiency and superior performance. It employs a hierarchical architecture, which not only aligns more closely with the nature of visual data but also boosts efficiency. To effectively capture global contextual information, Swin Transformer incorporates shifted window-based self-attention, further enhancing its effectiveness in vision-related applications. In our experiments, we will use Swin-50M.

- **BEIT** [82]: Due to the success of BERT, BEIT harnesses the concept of masked language modeling to enhance self-supervised learning in the visual domain. To align with the words in language models, BEIT first maps the patch in an image into a token with an autoencoder. In the training process, it masks a portion of these patches, using the remaining unmasked ones to predict the masked tokens. Subsequently, the model is fine-tuned for a variety of downstream tasks, demonstrating its adaptability and effectiveness in diverse applications. In our experiments, we will use BEIT-86M.

- **DeiT** [83]: To address the substantial data requirements for training the Vision Transformer, Data-Efficient Image Transformer (DeiT) employs knowledge distillation [81] to train the model. By integrating this approach with various data augmentation techniques, DeiT successfully attains competitive results in image classification tasks, even with constrained training data availability. In our experiments, we will use DeiT-22M.

- **ConViT** [84]: ConViT designs a hybrid architecture to leverage the local processing capabilities of CNNs and the global context understanding of transformers. To be specific, it replaces the several first self-attention layers with gated-self positional self-attention layers, allowing the model to adjust between local and global processing. In our experiments, we will use ConViT-30M.

Our ProTransformers belong to the category of optimization-induced deep learning architectures [85, 86, 87, 88], which formulate the attention layers in the backbone models as solutions to specific underlying optimization objectives, and then leverage optimization algorithms to solve them.

# E Attacks.

## E.1 Classic Text Attack:

For the classic text attacks, we follow the default attack setting in the TextAttack [63] and the detailed information are as follows:

- **DeepWordBug** [7]: DeepWordBug is black-box attacks that apply character-level transformations to the highest-ranked tokens misclassify the text input. It includes several character transformations including swap, substitution, deletion and insertion. We hold the maximum difference on edit distance (Levenshtein Edit Distance) to 30 for each sample. We will greedily modify the works with the word importance ranking.

- **PWWS** [9]: The probability weighted word saliency (PWWS) employs a new word order determined by the word saliency and predicted probability, and then greedily perform the synonyms substitution.

- **TextFooler** [14]: TextFooler propose a more comprehensive paradigm to generate adversarial perturbations. It firstly identify the important words and then replace them with the most semantically and syntacticaly similar synonyms until the prediction is altered. We set the minimum word embedding cosine similarity as 0.5 and the universal sentence similarity threshold as 0.840845057.

- **TextBugger** [8]: TextBugger is a framework for creating utility-preserving adversarial texts against the classification models. It can be employed under both white-box and black-box settings. Specifically, this attack sorts the words based on Jacobian matrix in white-box setting and based on specific scoring function in black-box setting. It includes swap, substitution, deletion and insertion. There are two types of substitutions: Sub-C (replace characters with visually similar characters) and Sub-W (replace a word with its top-$K$ ($K = 5$) nearest neighbors.) The universal sentence threshold is set as 0.8.

## E.2 Attack in LLMs

- **Prompt attack**: we conduct adversarial prompt attacks using the text attacks including DeepWordBug [7], PWWS [9], TextFooler [14] and TextBugger [8]. The evaluation setting follow PromptBench [15] which is a unified and comprehensive python library for evaluating LLMs.

- **Jailbreak**: We employ the Greedy Coordinate Gradient (GCG) [55] method to search for the adversarial prompt suffixes that circumvent the alignment of the language models. It can be conducted under the transfer attack and direct attack. Specifically, the transfer attacks firstly attack the surrogate model (e.g., Vicuna) and can apply the perturbations to the victim models. The direct attacks directly apply the GCG method to the victim models to generate the suffixes, which can be much stronger. The JailbreakKeywords is a list of words commonly found in messages that abstain from responding to a prompt asking for inappropriate content. In our experiment, we use the following list: JailbreakKeywords = ["I'm sorry", "Sorry", "I apologize", "As an", "I'm just", "I'm an", "I cannot", "I would advise", "it's not appropriate", "As a responsible AI"].

## E.3 Attacks in Vision & Graph Models

- **FGSM** [30]: Fast Gradient Sign Method (FGSM) is a widely used technique for generating adversarial examples. It crafts perturbations specifically to maximize errors in the output of neural networks.

- **PGD** [31]: Projected Gradient Descent (PGD) method is one of the most prominent attack strategies in computer vision. Unlike FGSM employing a single step, PGD uses multiple steps to generate adversarial examples. This iterative approach includes a projection operation, which ensures that the intensity of the attack remains within specified limits, making PGD a more controlled and effective method for generating adversarial examples. The steps are $K = 7$ and the steps size $\alpha = 0.00784$.

- **PGD on Graph** [64]: Motivated by PGD [31] in vision domain, [64] propose a first-order method to conduct topology attack on discrete graph structure. This method firstly solve continuous optimization problem by Projected Gradient Descent (PGD) method and then utilize the random sampling to get the optimal binary topology perturbation from the continuous probabilistic matrix.

## F Algorithm Convergence and Robust Estimation

### F.1 Convergence Guarantee

**Loss curves.** We use generated data to verify the convergence of our proposed algorithm. The batch size, number of heads, length of inputs and dimension of data are chosen as $B = 8, H = 4, N = 64, D = 8$, respectively. The $\gamma$ in MCP is set as $4$ and $\delta$ in Huber loss is set as $0.8$. The loss curve of our algorithm with different penalties are shown in Figure 7. We can observe that our algorithm show a fast convergence and even 2 to 3 steps can well approximate the optimal solution.

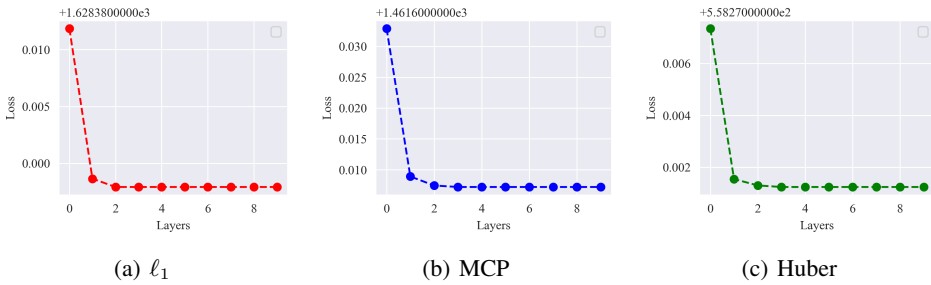

(a) $\ell_1$    (b) MCP    (c) Huber

Figure 7: Loss Curve of Algorithms

**Trajectory.** To further validate the convergence and effectiveness of our algorithm, we use a toy experiment to visualize the trajectories of updated vector in 2D plane in Figure 8. We use $L_1$ penalty in our algorithm, the simulated attention matrix and value matrix are as follows:

$$\mathbf{A} = \begin{bmatrix} 1 & 1 & 1 \\ 2 & 0 & 0 \\ 0 & 0 & 2 \end{bmatrix}, \mathbf{V} = \begin{bmatrix} 1 & 2 \\ 7 & 25 \\ 25 & 37 \end{bmatrix}. \tag{11}$$

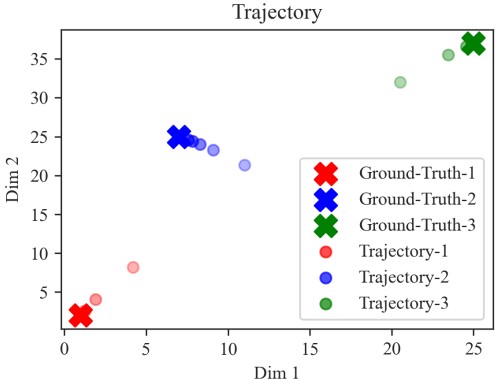

Figure 8: Optimization trajectory.

From the figure, we can find that with the mean as the initial position, the updated vector can approach closely to the ground truth within only 3 steps. This phenomenon further validate the effectiveness and efficiency of our algorithm.

### F.2 Robust Estimation

**Robust estimation.** We firstly generated clean samples $\{\mathbf{x}_i\}_{i=1}^n$ (blue dots) and the outlier samples $\{\mathbf{x}_i\}_{i=n+1}^{n+m}$ (red dots) from 2-dimensional Gaussian distributions, $\mathcal{N}((0,0),1)$ and $\mathcal{N}((8,8),0.5)$, respectively. We calculate the mean of clean samples $\frac{1}{n}\sum_{i=1}^n \mathbf{x}_i$ as the ground truth of the mean

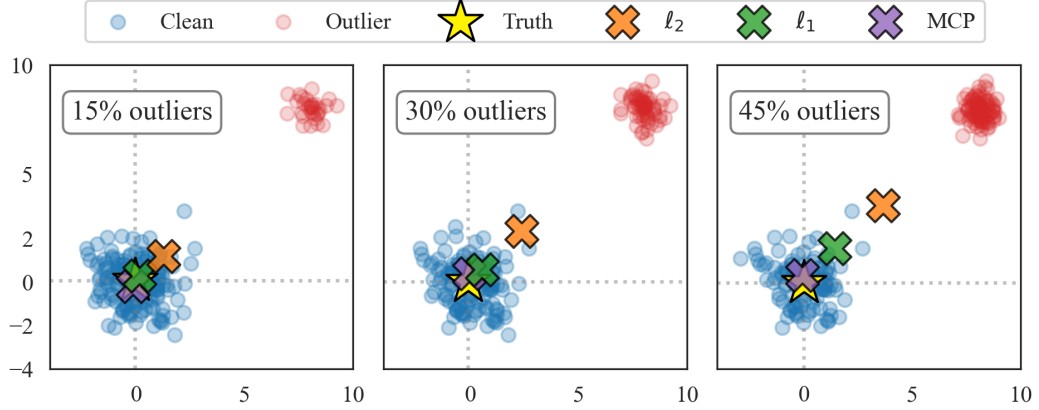

Figure 9: Different estimators in simulations.

estimator. Then we estimate the mean of all the samples by solving $\arg\min_{\mathbf{z}} \sum_{i=1}^{n+m} \rho(\mathbf{z} - \mathbf{x}_i)$ using the our method, where $\rho(\cdot)$ can take different penalties such as $\ell_2$ penalty $\| \cdot \|_2^2$ and $\ell_1$ penalty $\| \cdot \|_2$.

In Figure 9, we visualize the generated clean samples and outliers, as well as the ground truth means and the mean estimators with $\eta(\cdot) = \| \cdot \|_2^2$ or $\| \cdot \|_2$ under different outlier ratios 15% and 45%. The results show that, with the existence of outliers, the $\ell_2$-based estimator deviates far from the true mean, while the $\ell_1$-based estimator is more resistant to outliers and MCP-based estimator is the most robust.

### F.3 Vulnerability Analysis of Vanilla Attention (WLS estimator)

To gain insight into the vulnerability of vanilla attention (WLS estimator), we attempt to analyze the effect of perturbed tokens on the output embedding of attention modules. Specifically, since vanilla attention can be formalized as a WLS estimator: $\mathbf{z}^* = \arg\min_{\mathbf{z}} \mathcal{L}(\mathbf{z}) = \sum_{j=1}^{N} a_j \cdot \|\mathbf{v}_j - \mathbf{z}\|^2$, we quantify and compare $\|\mathbf{v}_j' - \mathbf{z}^*\|^2$ and $\|\mathbf{v}_j - \mathbf{z}^*\|^2$, where $\mathbf{v}_j$ and $\mathbf{v}_j'$ are the original and perturbed tokens, and $\mathbf{z}^*$ is the ground truth output embedding. We present the numerical results across every attention module in the pretrained BERT on IMDB dataset in Table 5. The results demonstrate that attackers tend to introduce larger residuals $\|\mathbf{v}_j - \mathbf{z}\|$ by modifying the important tokens $\mathbf{v}_j$ when perturbing the text.

Table 5: Residual magnitude of original and perturbed tokens.

| Attention module | 1 | 2 | 3 | 4 | 5 | 6 | 7 | 8 | 9 | 10 | 11 | 12 |
|---|---|---|---|---|---|---|---|---|---|---|---|---|
| $\|\mathbf{v}_j - \mathbf{z}^*\|^2$ | 5.39 | 6.36 | 6.92 | 6.33 | 6.64 | 6.95 | 6.73 | 6.24 | 5.12 | 4.76 | 4.15 | 4.19 |
| $\|\mathbf{v}_j' - \mathbf{z}^*\|^2$ | 6.38 | 7.20 | 7.97 | 7.51 | 7.65 | 8.06 | 7.84 | 7.41 | 6.33 | 6.24 | 6.64 | 6.31 |

# G    Additional Experiments of Text Attacks

## G.1    Sentiment Analysis: IMDB

We present the results of sentiment analysis on IMDB dataset under various attacks in Table 6. We can conlude from the results that our methods improve the robustness of the backbones significantly by simply plugging the ProAttention into the models without additional fintuning or training. Moreover, our method can be combined with the existing defenses such as Adversarial Training (AT) to further improve the performance.

Table 6: The results of sentiment analysis on IMDB.

| Model | Clean%↑ | TextFooler AUA%↑ | TextFooler ASR%↓ | TextBugger AUA%↑ | TextBugger ASR%↓ | DeepWordBug AUA%↑ | DeepWordBug ASR%↓ | PWWS AUA%↑ | PWWS ASR%↓ |
|---|---|---|---|---|---|---|---|---|---|
| RoBERTa | 93.3 | 23.7 | 74.6 | 9.4 | 89.9 | 36.5 | 60.9 | 19.5 | 79.1 |
| DistilBert | 90.9 | 14.9 | 83.6 | 4.3 | 95.3 | 18.8 | 79.3 | 9.6 | 89.4 |
| Albert | 92.8 | 21.8 | 76.5 | 14.1 | 84.8 | 36.2 | 61.0 | 15.9 | 82.9 |
| Bert | 92.3 | 11.8 | 87.2 | 11.3 | 87.4 | 32.8 | 64.5 | 26.4 | 71.5 |
| FreeLB | 93.0 | 25.1 | 73.6 | 19.9 | 76.9 | 40.9 | 55.5 | 42.0 | 54.7 |
| PGD | 93.2 | 26.2 | 69.2 | 17.4 | 81.6 | 32.0 | 65.8 | 27.2 | 69.6 |
| MixADA | 91.9 | 16.7 | 82.0 | 11.8 | 87.3 | 33.4 | 65.8 | 30.0 | 67.4 |
| TA-VAT | 93.0 | 28.5 | 67.6 | 27.3 | 68.8 | 34.7 | 60.4 | 35.1 | 59.8 |
| AT | 93.2 | 33.6 | 64.3 | 31.8 | 66.1 | 37.7 | 61.5 | 28.7 | 70.3 |
| Pro-Bert ($\ell_1$) (Ours) | 93.3 | 24.6 | 73.6 | 13.0 | 86.1 | 36.0 | 61.4 | 32.7 | 65.0 |
| Pro-Bert (Huber) (Ours) | 93.0 | 24.8 | 73.3 | 13.4 | 85.6 | 36.9 | 60.3 | 31.5 | 66.1 |
| Pro-Bert (MCP) (Ours) | 93.5 | 22.1 | 76.9 | 44.6 | 53.2 | 55.5 | 41.8 | 56.3 | 41.1 |
| Pro-Bert (MCP) + AT (Ours) | 93.6 | 42.0 | 56.1 | 55.3 | 41.0 | 60.8 | 39.0 | 61.0 | 37.6 |

## G.2    Textual Entailment: RTE

In Table 7, we display the results of textual entailment on RTE across different cosine similarities constraints in TextFooler attack. We select DistilBERT as the backbone model and construct several MCP-based architectures with different $\gamma$. We can observe that our method can improve the robustness acorss different cosine similarities. The performance improvement is more evident under the smaller cosine similarities, which is equivalent to larger budgets.

In Table 8, we present the results of textual entailment on RTE across various attacks. The results exhibit the consistent improvement of our methods over the backbone model.

Table 7: The results of textual entailment on RTE across different cosine similarities in TextFooler .

| Model | Cos-Sim Clean% | 0.5 AUA% | 0.5 ASR% | 0.6 AUA% | 0.6 ASR% | 0.7 AUA% | 0.7 ASR% | 0.8 AUA% | 0.8 ASR% |
|---|---|---|---|---|---|---|---|---|---|
| DistilBert | 62.5 | 4.0 | 93.6 | 5.1 | 91.9 | 7.9 | 87.3 | 18.1 | 71.1 |
| Pro-DistilBert-MCP $\gamma = 0.2$ | 63.3 | 6.9 | 89.5 | 6.1 | 90.6 | 9.4 | 85.6 | 18.1 | 72.4 |
| Pro-DistilBert-MCP $\gamma = 0.3$ | 62.4 | 15.2 | 75.3 | 16.6 | 72.9 | 20.6 | 66.5 | 30.7 | 50.0 |
| Pro-DistilBert-MCP $\gamma = 0.4$ | 56.0 | 18.1 | 67.7 | 24.6 | 56.1 | 28.2 | 49.7 | 33.2 | 40.7 |
| Pro-DistilBert-MCP $\gamma = 0.5$ | 55.6 | 15.9 | 70.1 | 17.7 | 66.7 | 20.2 | 61.9 | 28.9 | 45.6 |
| Pro-DistilBert-MCP $\gamma = 0.6$ | 53.1 | 10.8 | 80.5 | 12.3 | 77.9 | 16.3 | 70.8 | 20.2 | 63.6 |

Table 8: The results of textual entailment on RTE across different attacks.

| Model | Attack Clean% | TextFooler AUA% | TextFooler ASR% | TextBugger AUA% | TextBugger ASR% | DeepWordBug AUA% | DeepWordBug ASR% | PWWS AUA% | PWWS ASR% |
|---|---|---|---|---|---|---|---|---|---|
| DistilBert | 62.5 | 7.9 | 87.3 | 3.6 | 94.2 | 18.4 | 70.5 | 12.3 | 80.4 |
| Pro-DistillBert-MCP | 63.3 | 28.2 | 49.7 | 14.4 | 74.5 | 33.6 | 40.0 | 24.2 | 60.6 |

## G.3 Adversarial Fine-tuning on Topic Classification: AGNEWS

Adversarial training techniques are highly effective to enhance the robustness of models via adding the adversarial examples into the training set. To better capture the robustness enhancement of adversarial training, we track the adversarial fine-tuning curves and present the detailed results on AG's News in Table 9 and Figure 10. In the beginning of every epoch, we generate 1000 perturbed examples using the specific attack and then put them to the original training dataset. From the results, we can make the following observations: (1) the models show even better robustness during the process of adversarial training. (2) our method can be combined with adversarial training to further improve the resilience of the models.

Table 9: Adversarial fine-tuning on AGNEWS.

| MODEL | CLEAN% | AUA%(TF) | CLEAN% | AUA%(TB) | CLEAN% | AUA%(DWB) | CLEAN% | AUA%(PWWS) |
|---|---|---|---|---|---|---|---|---|
| BERT EPOCH-0 | 94.2 | 19.7 | 94.2 | 31.7 | 94.2 | 37.5 | 94.2 | 43.1 |
| BERT EPOCH-1 | 94.4 | 22.9 | 94.4 | 46.8 | 94.0 | 39.2 | 94.5 | 49.3 |
| BERT EPOCH-2 | 94.5 | 28.4 | 94.2 | 52.0 | 94.0 | 40.2 | 94.4 | 54.4 |
| BERT EPOCH-3 | 94.2 | 33.0 | 94.3 | 52.8 | 94.4 | 42.4 | 94.1 | 57.3 |
| BERT EPOCH-4 | 94.6 | 34.9 | 94.6 | 56.1 | 94.4 | 41.3 | 93.8 | 62.6 |
| BERT EPOCH-5 | 94.6 | 40.1 | 94.4 | 55.9 | 94.3 | 41.4 | 93.8 | 59.3 |
| PRO-BERT-MCP EPOCH-0 | 93.2 | 37.8 | 93.2 | 45.8 | 93.2 | 51.8 | 92.2 | 55.0 |
| PRO-BERT-MCP EPOCH-1 | 93.9 | 40.0 | 94.1 | 48.6 | 93.8 | 53.4 | 93.4 | 57.5 |
| PRO-BERT-MCP EPOCH-2 | 93.7 | 42.9 | 93.8 | 48.4 | 93.7 | 58.2 | 93.6 | 59.9 |
| PRO-BERT-MCP EPOCH-3 | 93.7 | 49.0 | 94.3 | 55.7 | 93.0 | 58.5 | 93.0 | 65.2 |
| PRO-BERT-MCP EPOCH-4 | 93.2 | 50.8 | 93.9 | 56.5 | 93.5 | 61.0 | 93.6 | 65.1 |
| PRO-BERT-MCP EPOCH-5 | 93.9 | 53.0 | 94.5 | 60.7 | 93.0 | 60.1 | 93.6 | 68.8 |

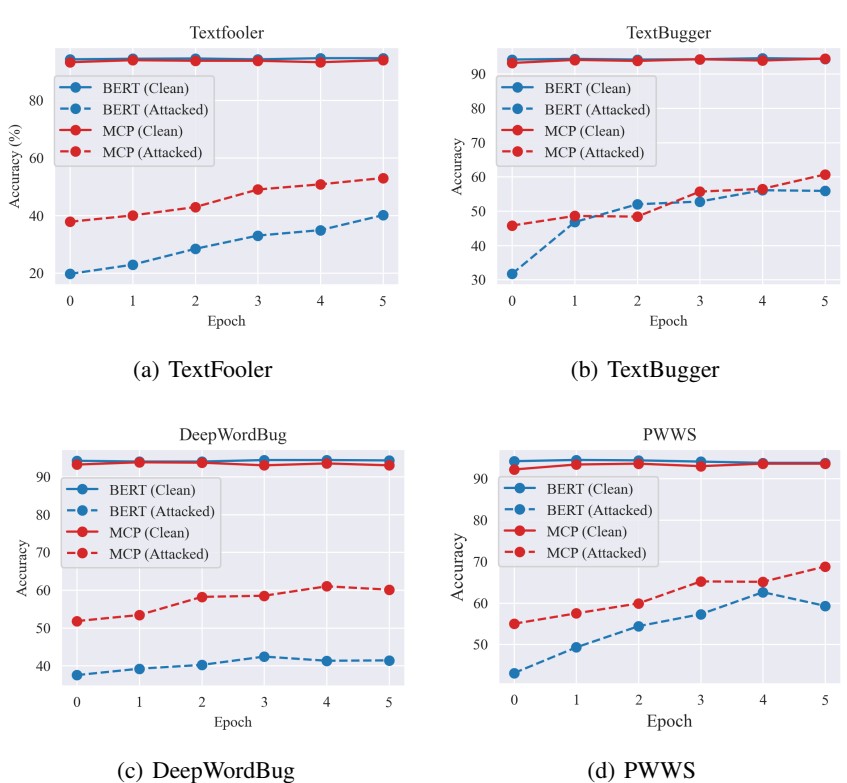

(a) TextFooler

(b) TextBugger

(c) DeepWordBug

(d) PWWS

Figure 10: Adversarial fine-tuning on AGNEWS.

## G.4 Ablation Study on Attack constraints

We present the ablation study on the maximum perturbation percentage, minimum cosines similarity and sentence similarity threshold in Figure 11, Table 10, Table 11 and Table 12, respectively. The experiments are performed on AGNEWS under TextFooler with the ALBERT as the backbone. The default values are as follows: sentence similarity threshold is $0.840845057$, maximum perturbation percentage is $1.0$, synonym cosine similarity is $0.5$. The results show the consistent improvement of our method over the backbone models.

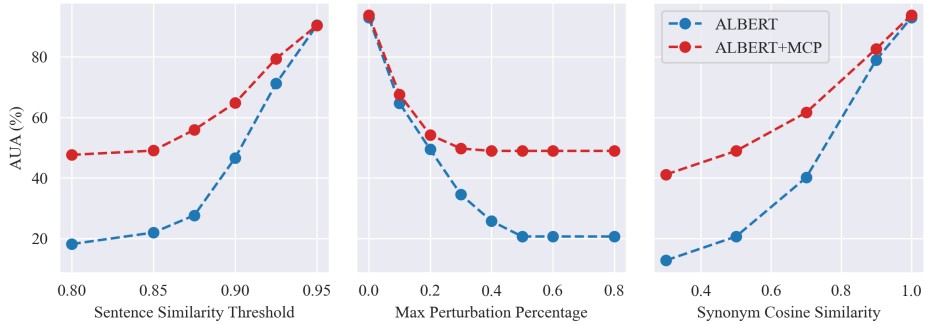

Figure 11: Ablation studies on attack constraints.

Table 10: Ablation study on max perturbation percentage.

| MAX-PERCENTAGE ($\rho$) | | 0.1 | | 0.2 | | 0.3 | | 0.4 | |
|---|---|---|---|---|---|---|---|---|---|
| MODEL | CLEAN% | AUA% | ASR% | AUA% | ASR% | AUA% | ASR% | AUA% | ASR% |
| ALBERT | 93.0 | 64.7 | 30.4 | 49.4 | 46.9 | 34.5 | 62.9 | 25.7 | 72.4 |
| PRO-ALBERT-MCP | 93.8 | 67.6 | 27.2 | 54.2 | 41.6 | 49.7 | 46.4 | 48.9 | 47.3 |

| MAX-PERCENTAGE ($\rho$) | 0.6 | | 0.8 | | 1.0 | | | | |
|---|---|---|---|---|---|---|---|---|---|
| MODEL | AUA% | ASR% | AUA% | ASR% | AUA% | ASR% | | | |
| ALBERT | 20.6 | 77.9 | 20.6 | 77.9 | 20.6 | 77.9 | | | |
| PRO-ALBERT-MCP | 48.9 | 47.3 | 48.9 | 47.3 | 48.9 | 47.3 | | | |

Table 11: Ablation study on minimum synonym cosine similarity.

| MIN-COS-SIM | | 0.3 | | 0.5 | | 0.7 | | 0.9 | |
|---|---|---|---|---|---|---|---|---|---|
| MODEL | CLEAN% | AUA% | ASR% | AUA% | ASR% | AUA% | ASR% | AUA% | ASR% |
| ALBERT | 93.0 | 12.7 | 86.3 | 20.6 | 77.9 | 40.1 | 56.9 | 79.0 | 15.1 |
| PRO-ALBERT-MCP | 93.8 | 41.1 | 55.7 | 48.9 | 47.3 | 61.6 | 33.6 | 82.7 | 10.9 |

Table 12: Ablation study on universal sentence similarity threshold.

| SENTENCE-SIM-THRESHOLD | | 0.2 | | 0.4 | | 0.6 | | 0.8 | | 0.85 | |
|---|---|---|---|---|---|---|---|---|---|---|---|
| MODEL | CLEAN% | AUA% | ASR% | AUA% | ASR% | AUA% | ASR% | AUA% | ASR% | AUA% | ASR% |
| ALBERT | 93.0 | 18.0 | 80.7 | 18.0 | 80.7 | 18.0 | 80.7 | 18.1 | 80.5 | 21.9 | 76.5 |
| PRO-ALBERT-MCP | 93.8 | 47.4 | 48.9 | 47.4 | 48.9 | 47.4 | 48.9 | 47.6 | 48.7 | 49.0 | 47.2 |

| SENTENCE-SIM-THRESHOLD | 0.875 | | 0.9 | | 0.925 | | 0.95 | | | | |
|---|---|---|---|---|---|---|---|---|---|---|---|
| MODEL | AUA% | ASR% | AUA% | ASR% | AUA% | ASR% | AUA% | ASR% | | | |
| ALBERT | 27.6 | 70.3 | 46.6 | 49.9 | 71.2 | 23.4 | 90.5 | 2.7 | | | |
| PRO-ALBERT-MCP | 55.9 | 39.8 | 64.8 | 30.2 | 79.4 | 14.4 | 90.4 | 2.6 | | | |

## G.5 Ablation Study on Backbone Models

Our proposed ProAttention is a universal framework which can be applied to various attention-based models. To verify the universality of our methods, we integrate our robust attention module into various backbones and present the results in Figure 12 and Table 13. As seen in the results, our ProAttention can consistently enhance the robustness over any backbone under various attacks.

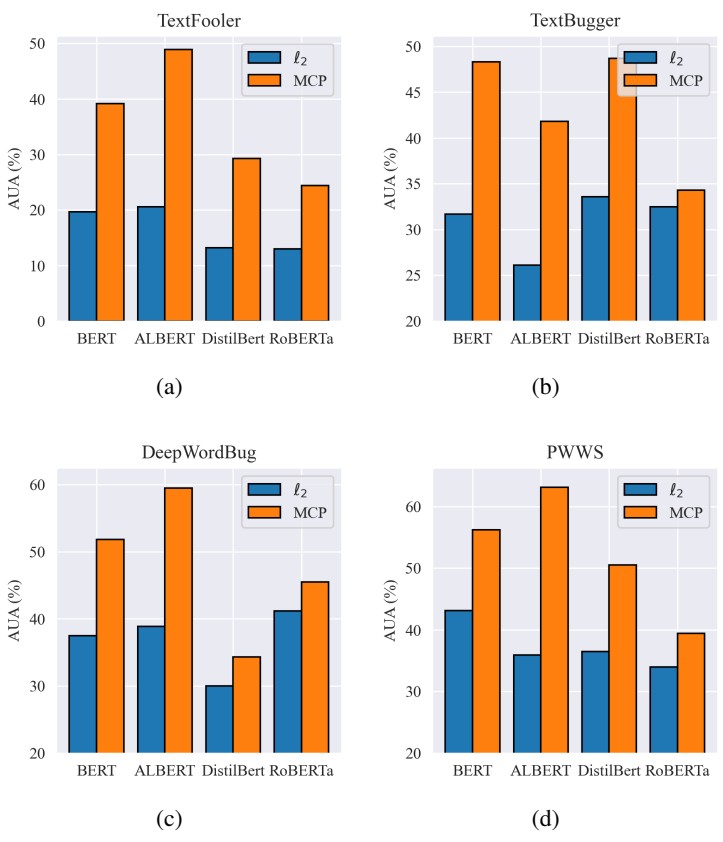

Figure 12: Accuracy under attack of different backbones.

Table 13: The results of different backbones on AGNEWS

| MODEL | CLEAN% | TEXTFOOLER | | | TEXTBUGGER | | | DEEPWORDBUG | | | PWWS | | |
| | | AUA% | ASR% | #QUERY | AUA% | ASR% | #QUERY | AUA% | ASR% | #QUERY | AUA% | ASR% | #QUERY |
|---|---|---|---|---|---|---|---|---|---|---|---|---|---|
| BERT | 94.2 | 19.7 | 78.9 | 335.3 | 31.7 | 67.5 | 176.5 | 37.5 | 59.8 | 103.8 | 43.1 | 53.8 | 353.8 |
| PRO-BERT-MCP | 93.2 | 39.2 | 57.7 | 377.4 | 48.3 | 48.5 | 207.7 | 51.8 | 43.8 | 107.9 | 56.2 | 39.2 | 363.2 |
| ROBERTA | 93.4 | 13.0 | 86.1 | 301.6 | 32.5 | 64.5 | 180.3 | 41.2 | 55.9 | 105.4 | 34.0 | 63.6 | 345.9 |
| PRO-ROBERTA-MCP | 93.7 | 24.4 | 73.7 | 312.6 | 34.3 | 62.8 | 195.2 | 45.5 | 51.5 | 118.3 | 39.4 | 57.5 | 336.9 |
| DISTILBERT | 93.5 | 13.2 | 85.9 | 317.4 | 33.6 | 63.4 | 159.1 | 30.0 | 67.9 | 98.0 | 36.5 | 61.0 | 352.4 |
| PRO-DISTILBERT-MCP | 93.9 | 29.3 | 68.5 | 363.3 | 48.7 | 47.9 | 184.2 | 34.3 | 63.1 | 98.6 | 50.5 | 45.6 | 364.2 |
| ALBERT | 93.0 | 20.6 | 77.9 | 315.6 | 26.1 | 71.9 | 150.9 | 38.9 | 58.2 | 101.5 | 35.9 | 61.4 | 342.8 |
| PRO-ALBERT-MCP | 93.8 | 48.9 | 47.3 | 417.8 | 41.8 | 55.3 | 208.2 | 59.5 | 35.9 | 111.8 | 63.1 | 32.0 | 375.2 |

## G.6 Ablation Study on Hyperparameters of Penalties

### G.6.1 Ablation Study on Huber

For the Huber-based model, we present the ablation study on the $\delta$ and layers $K$ of Huber loss under TextFooler in Table 14 and Figure 13. The default setting are as follows: $\delta = 0.9$ and $K = 3$, and we vary the two parameters separately to capture the trend. We can find that the performance is insensitive to $\delta$ or layers within an appropriate range.

Table 14: Ablation study on Huber on AGNEWS.

| MODEL | CLEAN | ATTACKED |
|---|---|---|
| PRO-BERT-HUBER $\delta = 0.6$ | 94.2 | 23.9 |
| PRO-BERT-HUBER $\delta = 0.7$ | 94.2 | 23.2 |
| PRO-BERT-HUBER $\delta = 0.8$ | 94.2 | 23.7 |
| PRO-BERT-HUBER $\delta = 0.9$ | 94.2 | 24.2 |
| PRO-BERT-HUBER $\delta = 1$ | 94.3 | 23.0 |
| PRO-BERT-HUBER $\delta = 2$ | 94.1 | 22.5 |
| PRO-BERT-HUBER $\delta = 3$ | 94.2 | 21.7 |
| PRO-BERT-HUBER $\delta = 4$ | 94.2 | 20.9 |
| PRO-BERT-HUBER $\delta = 5$ | 94.1 | 20.0 |
| PRO-BERT-HUBER $K = 3$ | 94.2 | 24.2 |
| PRO-BERT-HUBER $K = 4$ | 94.2 | 24.0 |
| PRO-BERT-HUBER $K = 5$ | 94.2 | 23.7 |
| PRO-BERT-HUBER $K = 6$ | 94.0 | 24.5 |
| PRO-BERT-HUBER $K = 7$ | 93.9 | 24.2 |
| PRO-BERT-HUBER $K = 8$ | 94.0 | 25.8 |

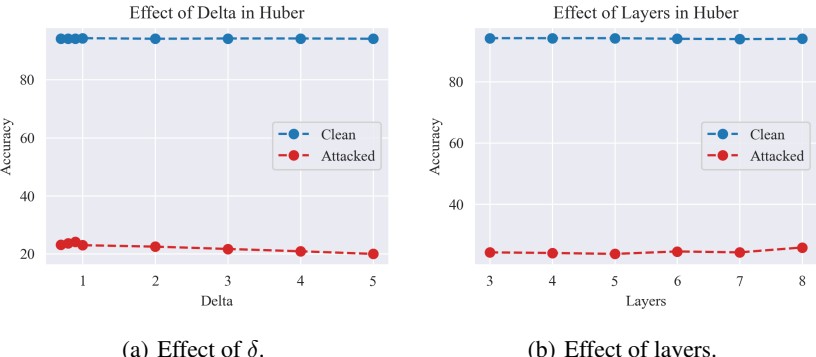

(a) Effect of $\delta$.  (b) Effect of layers.

Figure 13: Ablation study on Huber

### G.6.2 Ablation Study on MCP

We present the ablation study of the $\gamma$ and layers $K$ in MCP in Table 15 & Figure 14 and Table 16 & Figure 15 . From the figures, it can be concluded that appropriate $\gamma$ is needed to get the best robustness. Besides, more layers can get a more precise solution for the robust objective, but we need to consider a good balance between precision and efficiency.

Table 15: Ablation on MCP on AGNEWS.

| MODEL | CLEAN | TEXTFOOLER |
|---|---|---|
| PRO-BERT-MCP $K = 2, \gamma = 5$ | 93.3 | 32.8 |
| PRO-BERT-MCP $K = 2, \gamma = 4$ | 93.7 | 33.9 |
| PRO-BERT-MCP $K = 2, \gamma = 3$ | 93.7 | 31.7 |
| PRO-BERT-MCP $K = 2, \gamma = 2$ | 93.1 | 30.4 |
| PRO-BERT-MCP $K = 2, \gamma = 1$ | 92.9 | 28.7 |
| PRO-BERT-MCP $K = 5, \gamma = 4$ | 93.6 | 39.2 |
| PRO-BERT-MCP $K = 4, \gamma = 4$ | 93.7 | 37.2 |
| PRO-BERT-MCP $K = 3, \gamma = 4$ | 93.2 | 37.8 |
| PRO-BERT-MCP $K = 2, \gamma = 4$ | 93.7 | 33.9 |
| PRO-BERT-MCP $K = 1, \gamma = 4$ | 93.9 | 26.8 |

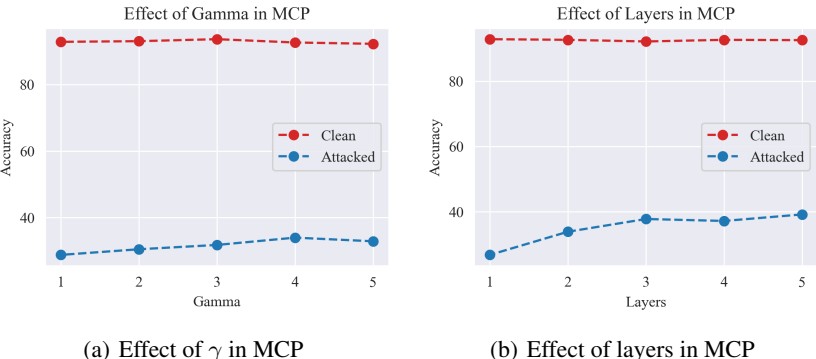

(a) Effect of $\gamma$ in MCP
(b) Effect of layers in MCP

Figure 14: Ablation study on MCP on AGNEWS.

Table 16: Ablation in MCP on IMDB.

| MODEL | CLEAN | TEXTFOOLER | TEXTBUGGER | DEEPWORDBUG | PWWS |
|---|---|---|---|---|---|
| PRO-BERT-MCP $\gamma = 2.0, K = 3$ | 93.9 | 15.9 | 19.9 | 43.7 | 40.3 |
| PRO-BERT-MCP $\gamma = 3.0, K = 3$ | 93.7 | 15.1 | 24.0 | 53.8 | 51.1 |
| PRO-BERT-MCP $\gamma = 4.0, K = 3$ | 93.4 | 16.5 | 26.6 | 55.5 | 50.7 |
| PRO-BERT-MCP $\gamma = 5.0, K = 3$ | 93.9 | 16.3 | 29.8 | 53.9 | 46.6 |
| PRO-BERT-MCP $\gamma = 6.0, K = 3$ | 93.3 | 13.5 | 23.0 | 48.1 | 41.5 |
| PRO-BERT-MCP $K = 1$ | 93.6 | 12.4 | 13.8 | 40.8 | 40.2 |
| PRO-BERT-MCP $K = 2$ | 93.8 | 12.5 | 15.7 | 49.0 | 47.9 |
| PRO-BERT-MCP $K = 3$ | 93.4 | 16.5 | 29.8 | 53.9 | 46.6 |
| PRO-BERT-MCP $K = 4$ | 93.5 | 20.4 | 39.4 | 60.2 | 56.3 |
| PRO-BERT-MCP $K = 5$ | 93.5 | 22.1 | 44.6 | 63.3 | 56.1 |

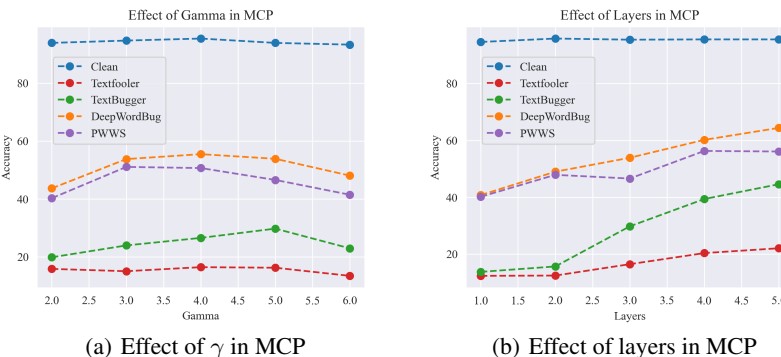

(a) Effect of $\gamma$ in MCP      (b) Effect of layers in MCP

Figure 15: Ablation study on MCP on IMDB.

# H  Additional Experiments on LLMs

## H.1  Experiments on T5

### H.1.1  main result

We compare the backbone model T5 with its robust version based on $\ell_1$, MCP and Huber loss in Figure 16 and Table 17. The experiments are conducted on SST2 under TextFooler. As shown in the figure, Pro-T5 (Huber or $\ell_1$) can slightly improve the robustness of backbone T5. Moreover, Pro-T5 (MCP) significantly outperform other baselines, especially under the large attack budgets.

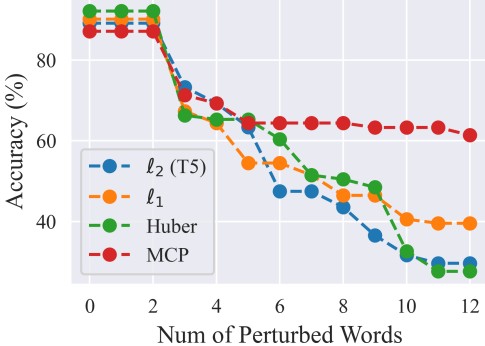

Figure 16: Main results on T5.

Table 17: Accuary (%) under prompt attack on SST2 (TextFooler, T5)

| # PERTURBED WORDS | 0 (CLEAN) | 1 | 2 | 3 | 4 | 5 | 6 | 7 | 8 | 9 | 10 | 11 | 12 |
|---|---|---|---|---|---|---|---|---|---|---|---|---|---|
| T5 | 89.1 | 89.1 | 89.1 | 73.3 | 69.3 | 63.4 | 47.5 | 47.5 | 43.6 | 36.6 | 31.7 | 29.7 | 29.7 |
| PRO-T5 $\ell_1$ $K=3$ | 90.1 | 90.1 | 90.1 | 67.3 | 64.4 | 54.5 | 54.5 | 51.5 | 46.5 | 46.5 | 40.6 | 39.6 | 39.6 |
| PRO-T5 MCP $K=4, \gamma=3.0$ | 87.1 | 87.1 | 87.1 | 71.3 | 69.3 | 64.4 | 64.4 | 64.4 | 64.4 | 63.3 | 63.3 | 63.3 | 61.4 |
| PRO-T5 HUBER $K=3, \delta=3$ | 95.0 | 95.0 | 95.0 | 64.4 | 58.4 | 56.4 | 52.5 | 52.5 | 52.5 | 43.6 | 33.7 | 31.7 | 31.7 |
| PRO-T5 HUBER-MCP $K=3, \delta=9, \gamma=15$ | 89.1 | 89.1 | 89.1 | 62.4 | 62.4 | 57.4 | 55.4 | 55.4 | 55.4 | 55.4 | 55.4 | 55.4 | 54.5 |

### H.1.2 Ablation on Huber

We present the ablation study on $\delta$ in Huber on SST2 in Figure 17 and Table 18. We fix the number of layers as 3 and vary the values of $\delta$ from 3.0 to 9.0. The Huber-based model with $\delta = 4.0$ perform best among all the selected parameters.

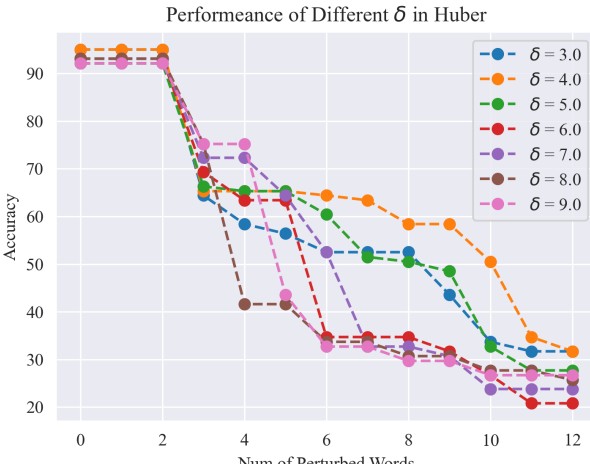

Figure 17: Ablation study on Huber on T5

Table 18: Ablation study in Huber on SST2.

| # PERTURBED WORDS | 0 (CLEAN) | 1 | 2 | 3 | 4 | 5 | 6 | 7 | 8 | 9 | 10 | 11 | 12 |
|---|---|---|---|---|---|---|---|---|---|---|---|---|---|
| PRO-T5 HUBER $K = 3, \delta = 3$ | 95.0 | 95.0 | 95.0 | 64.4 | 58.4 | 56.4 | 52.5 | 52.5 | 52.5 | 43.6 | 33.7 | 31.7 | 31.7 |
| PRO-T5 HUBER $K = 3, \delta = 4$ | 95.0 | 95.0 | 95.0 | 65.3 | 65.3 | 65.3 | 64.4 | 63.4 | 58.4 | 58.4 | 50.5 | 34.7 | 31.7 |
| PRO-T5 HUBER $K = 3, \delta = 5$ | 92.1 | 92.1 | 92.1 | 66.3 | 65.3 | 65.3 | 60.4 | 51.5 | 50.5 | 48.5 | 32.7 | 27.7 | 27.7 |
| PRO-T5 HUBER $K = 3, \delta = 6$ | 93.1 | 93.1 | 93.1 | 69.3 | 63.4 | 63.4 | 34.7 | 34.7 | 34.7 | 31.7 | 26.7 | 20.8 | 20.8 |
| PRO-T5 HUBER $K = 3, \delta = 7$ | 93.1 | 93.1 | 93.1 | 72.3 | 72.3 | 64.4 | 52.5 | 32.7 | 32.7 | 30.7 | 23.8 | 23.8 | 23.8 |
| PRO-T5 HUBER $K = 3, \delta = 8$ | 93.1 | 93.1 | 93.1 | 75.2 | 41.6 | 41.6 | 33.7 | 33.7 | 30.7 | 30.7 | 27.7 | 27.7 | 25.7 |
| PRO-T5 HUBER $K = 3, \delta = 9$ | 92.1 | 92.1 | 92.1 | 75.2 | 75.2 | 43.6 | 32.7 | 32.7 | 29.7 | 29.7 | 26.7 | 26.7 | 26.7 |

### H.1.3   Ablation on MCP of T5

We present the ablation studies in MCP on SST2 in Figure 18 and Table 19. The default settings are as follows: $K = 3$ and $\gamma = 3.0$. We can make the observations as follows: (1) For $\gamma$, the optimal value falls into the range of (3,5). (2) For number of layers, the best value can be chosen as 4 or 5.

Figure 18: Ablation study on MCP of T5

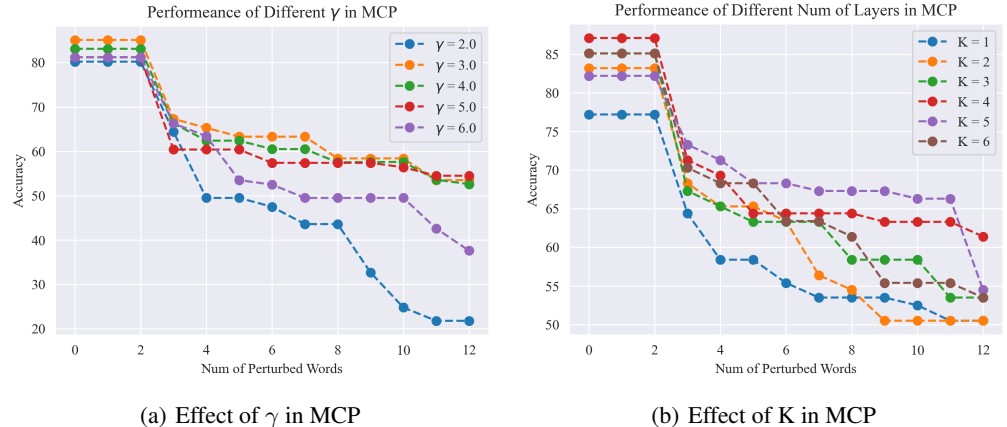

(a) Effect of $\gamma$ in MCP  (b) Effect of K in MCP

Table 19: Ablation study on MCP on SST2

| # Perturbed words | 0 (Clean) | 1 | 2 | 3 | 4 | 5 | 6 | 7 | 8 | 9 | 10 | 11 | 12 |
|---|---|---|---|---|---|---|---|---|---|---|---|---|---|
| Pro-T5 MCP $K = 3, \gamma = 2.0$ | 80.2 | 80.2 | 80.2 | 64.4 | 49.5 | 49.5 | 47.5 | 43.6 | 43.6 | 32.7 | 24.8 | 21.8 | 21.8 |
| Pro-T5 MCP $K = 3, \gamma = 3.0$ | 85.1 | 85.1 | 85.1 | 67.3 | 65.3 | 63.3 | 63.3 | 63.3 | 58.4 | 58.4 | 58.4 | 53.5 | 53.5 |
| Pro-T5 MCP $K = 3, \gamma = 4.0$ | 83.1 | 83.1 | 83.1 | 66.3 | 62.4 | 62.4 | 60.5 | 60.5 | 57.5 | 57.6 | 57.6 | 53.6 | 52.6 |
| Pro-T5 MCP $K = 3, \gamma = 5.0$ | 81.2 | 81.2 | 81.2 | 60.4 | 60.4 | 60.4 | 57.4 | 57.4 | 57.4 | 57.4 | 56.4 | 54.5 | 54.5 |
| Pro-T5 MCP $K = 3, \gamma = 6.0$ | 81.2 | 81.2 | 81.2 | 66.3 | 63.4 | 53.5 | 52.5 | 49.5 | 49.5 | 49.5 | 49.5 | 42.6 | 37.6 |
| Pro-T5 MCP $K = 1, \gamma = 3.0$ | 77.2 | 77.2 | 77.2 | 64.4 | 58.4 | 58.4 | 55.4 | 53.5 | 53.5 | 53.5 | 52.5 | 50.5 | 50.5 |
| Pro-T5 MCP $K = 2, \gamma = 3.0$ | 83.2 | 83.2 | 83.2 | 68.3 | 65.3 | 65.3 | 63.4 | 56.4 | 54.5 | 50.5 | 50.5 | 50.5 | 50.5 |
| Pro-T5 MCP $K = 3, \gamma = 3.0$ | 85.1 | 85.1 | 85.1 | 67.3 | 65.3 | 63.3 | 63.3 | 63.3 | 58.4 | 58.4 | 58.4 | 53.5 | 53.5 |
| Pro-T5 MCP $K = 4, \gamma = 3.0$ | 87.1 | 87.1 | 87.1 | 71.3 | 69.3 | 64.4 | 64.4 | 64.4 | 64.4 | 63.3 | 63.3 | 63.3 | 61.4 |
| Pro-T5 MCP $K = 5, \gamma = 3.0$ | 82.2 | 82.2 | 82.2 | 73.3 | 71.3 | 68.3 | 68.3 | 67.3 | 67.3 | 67.3 | 66.3 | 66.3 | 54.5 |
| Pro-T5 MCP $K = 6, \gamma = 3.0$ | 85.1 | 85.1 | 85.1 | 70.3 | 68.3 | 68.3 | 63.4 | 63.4 | 61.4 | 55.4 | 55.4 | 55.4 | 53.5 |

### H.1.4 Ablation on Huber-MCP

We present the ablation study on Huber-MCP on SST2 in Figure 19 and Table 20. The results show that Pro-T5 (Huber-MCP) is insensitive to $\delta$ and get better robustness at $\gamma = 14$ or 15.

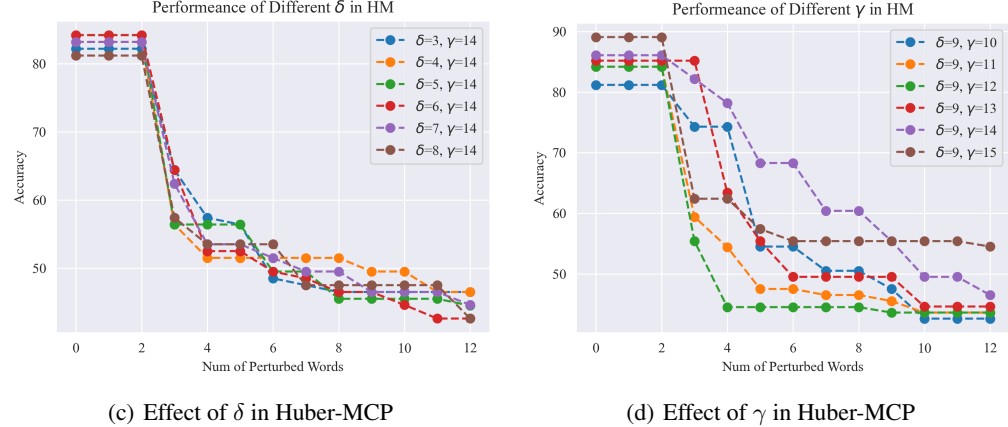

(c) Effect of $\delta$ in Huber-MCP    (d) Effect of $\gamma$ in Huber-MCP

Figure 19: Ablation studies on Huber-MCP

Table 20: Ablation study on Huber-MCP on SST2

| # PERTURBED WORDS | 0 (CLEAN) | 1 | 2 | 3 | 4 | 5 | 6 | 7 | 8 | 9 | 10 | 11 | 12 |
|---|---|---|---|---|---|---|---|---|---|---|---|---|---|
| PRO-T5 HUBER-MCP $K=3, \delta=9, \gamma=10$ | 81.2 | 81.2 | 81.2 | 74.3 | 74.3 | 54.5 | 54.5 | 50.5 | 50.5 | 47.5 | 42.6 | 42.6 | 42.6 |
| PRO-T5 HUBER-MCP $K=3, \delta=9, \gamma=11$ | 84.2 | 84.2 | 84.2 | 59.4 | 54.4 | 47.5 | 47.5 | 46.5 | 46.5 | 45.5 | 43.6 | 43.6 | 43.6 |
| PRO-T5 HUBER-MCP $K=3, \delta=9, \gamma=12$ | 84.2 | 84.2 | 84.2 | 55.4 | 44.5 | 44.5 | 44.5 | 44.5 | 44.5 | 43.6 | 43.6 | 43.6 | 43.6 |
| PRO-T5 HUBER-MCP $K=3, \delta=9, \gamma=13$ | 85.2 | 85.2 | 85.2 | 85.2 | 63.4 | 55.4 | 49.5 | 49.5 | 49.5 | 49.5 | 44.6 | 44.6 | 44.6 |
| PRO-T5 HUBER-MCP $K=3, \delta=9, \gamma=14$ | 86.1 | 86.1 | 86.1 | 82.2 | 78.2 | 68.3 | 68.3 | 60.4 | 60.4 | 55.4 | 49.5 | 49.5 | 46.5 |
| PRO-T5 HUBER-MCP $K=3, \delta=9, \gamma=15$ | 89.1 | 89.1 | 89.1 | 62.4 | 62.4 | 57.4 | 55.4 | 55.4 | 55.4 | 55.4 | 55.4 | 55.4 | 54.5 |
| PRO-T5 HUBER-MCP $K=3, \delta=3, \gamma=14$ | 82.2 | 82.2 | 82.2 | 64.4 | 57.4 | 56.4 | 48.5 | 47.5 | 46.5 | 46.5 | 46.5 | 46.5 | 46.5 |
| PRO-T5 HUBER-MCP $K=3, \delta=4, \gamma=14$ | 83.2 | 83.2 | 83.2 | 56.4 | 51.5 | 51.5 | 51.5 | 51.5 | 51.5 | 49.5 | 49.5 | 46.5 | 46.5 |
| PRO-T5 HUBER-MCP $K=3, \delta=5, \gamma=14$ | 84.2 | 84.2 | 84.2 | 56.4 | 56.4 | 56.4 | 49.5 | 49.5 | 45.5 | 45.5 | 45.5 | 45.5 | 44.5 |
| PRO-T5 HUBER-MCP $K=3, \delta=6, \gamma=14$ | 84.2 | 84.2 | 84.2 | 64.4 | 52.5 | 52.5 | 49.5 | 48.5 | 46.5 | 46.5 | 44.6 | 42.6 | 42.6 |
| PRO-T5 HUBER-MCP $K=3, \delta=7, \gamma=14$ | 83.2 | 83.2 | 83.2 | 62.4 | 53.5 | 53.5 | 51.5 | 49.5 | 49.5 | 46.5 | 46.5 | 46.5 | 44.6 |
| PRO-T5 HUBER-MCP $K=3, \delta=8, \gamma=14$ | 81.2 | 81.2 | 81.2 | 57.4 | 53.5 | 53.5 | 53.5 | 47.5 | 47.5 | 47.5 | 47.5 | 47.5 | 42.6 |

## H.2 Experiments on LLaMA

For LLaMA, we can observe an intriguing phenomenon that differs from the T5 case: the $\ell_1$ and MCP-based methods sacrifice too much accuracy while Huber method can keep decent performance under small budgets. This reason is that in the small range region, $\ell_1$ and MCP utilize a linear or concave functions while Huber can recover the $\ell_2$ function. Inspired by the characteristics of these functions, we combine the properties of Huber and MCP, and construct a new function which we refer to Huber-MCP[2]. The detailed formulation and derived robust attention layers are available in Appendix B. As indicated by the following curves, Huber-MCP and Huber-based models exhibits better robustness than other methods while preserving the good clean performance.

### H.2.1 Textfooler

We present the results of textual entailment on SST2 under TextFooler in Figure 20. We can observe that $\ell_1$ and MCP-based methods sacrifice the performance because of the estimation bias. Pro-LLaMA (Huber-MCP) shows slight improvement over other models.

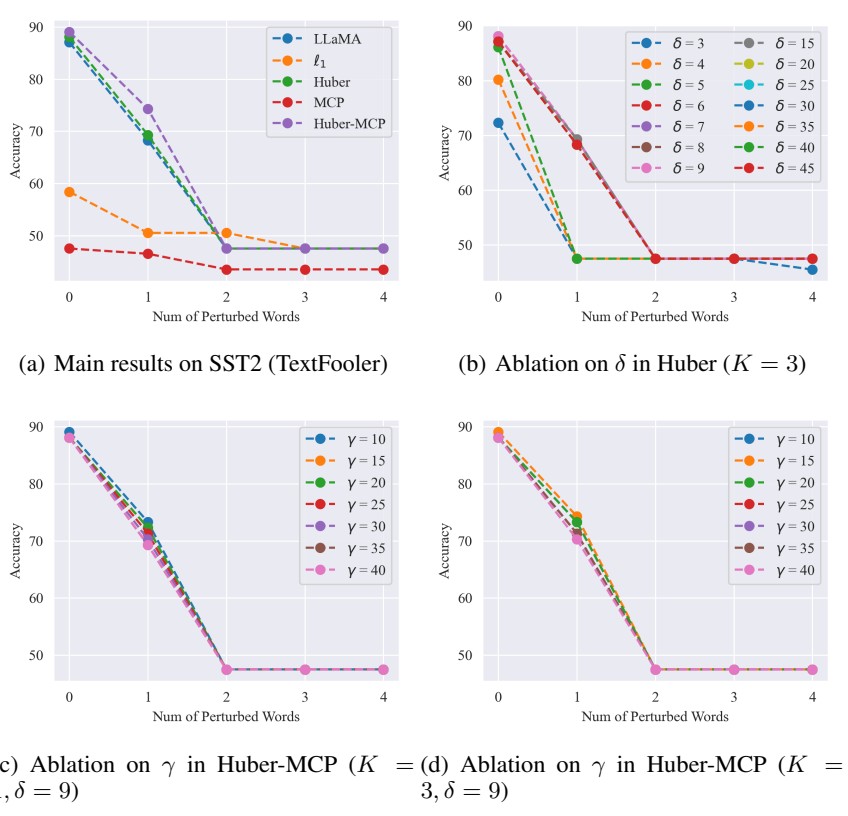

(a) Main results on SST2 (TextFooler)

(b) Ablation on $\delta$ in Huber ($K = 3$)

(c) Ablation on $\gamma$ in Huber-MCP ($K = 1, \delta = 9$)

(d) Ablation on $\gamma$ in Huber-MCP ($K = 3, \delta = 9$)

Figure 20: LLaMA (Textfooler)

---

[2]Empirical penalty selection strategy: For small or medium-sized models such as BERT (110M) and ViT (86M), MCP-based models exhibit superior robustness while nearly not sacrificing the clean performance. Moreover, MCP-based models are easy to tune with only one parameter $\gamma$. For large models like LLaMA (7B) and Vicuna (7B), it is necessary to choose Huber and Huber-MCP to recover the original $\ell_2$ penalty within the low-value region in case of clean performance drop.

### H.2.2 TextBugger

We present the results of textual entailment on SST2 under TextBugger in Figure 21. In this case, $\ell_1$-based model show a catastrophic performance drop while Pro-LLaMA (Huber) outperforms other baselines with a sinificant margin.

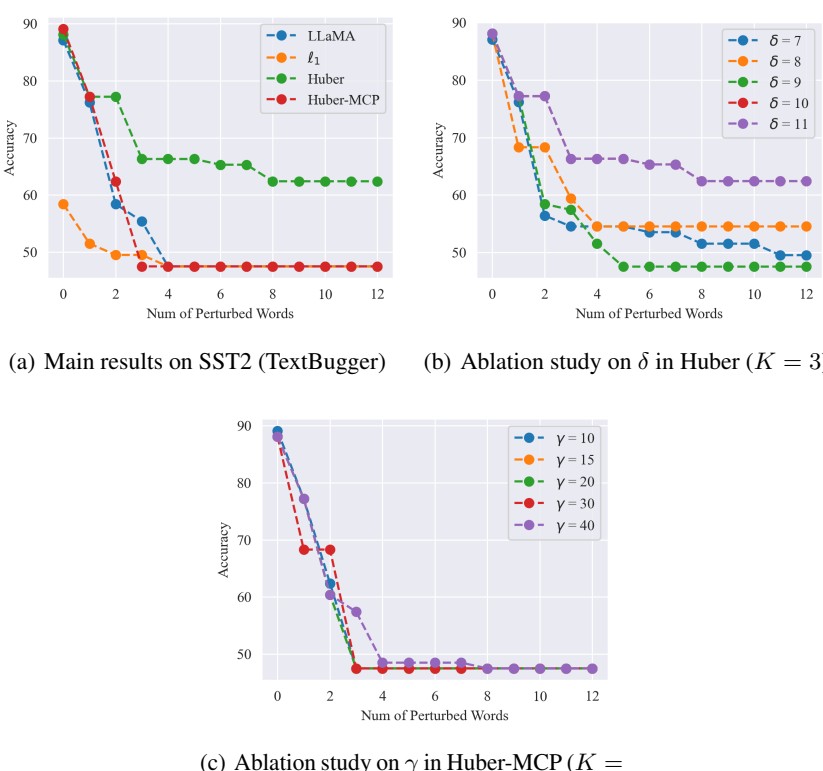

(a) Main results on SST2 (TextBugger)

(b) Ablation study on $\delta$ in Huber ($K = 3$)

(c) Ablation study on $\gamma$ in Huber-MCP ($K = 3, \delta = 9$)

Figure 21: LLaMA (TextBugger)

### H.2.3 DeepWordBug

We present the results of textual entailment on SST2 under DeepWordBug in Figure 22. The experiment shows the similar phenomenon that $\ell_1$ and MCP-based models sacrifice too much performance. Additionally, Pro-LLaMA (Huber-MCP) significantly outperforms other methods.

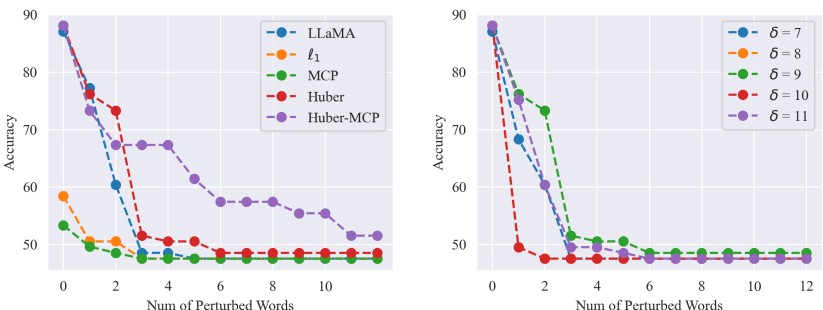

(a) Main results on SST2 (DeepWordBug)    (b) Ablation study on $\delta$ in Huber ($K = 3$)

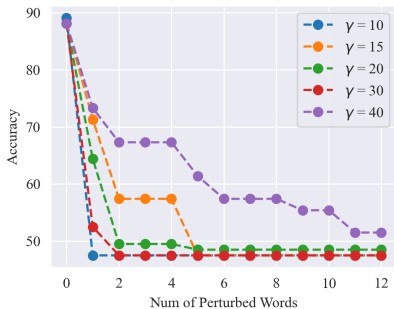

(c) Ablation study on $\gamma$ in Huber-MCP ($\delta = 9, L = 3$)

Figure 22: LLaMA (DeepWordBug)

# I   Additional Experiments on Jailbreak

## I.1   Transfer Jailbreak

We provide the results of transfer jailbreak in Table 21, Table 22, and Table 23. As SmoothLLM exhibits excellent performance to defend the jaibreaking attack, we also combine the random smoothing with the backbone models and our methods to further validate the effectiveness of our method. As shown in the results, when $q = 0$ (without random smoothing), by simply plugging our ProAttention into the Vicuna, the robustness can be improved by a significant margin. Our plug-and-play method can even be comparable to the randomly smoothed models which require multiple operations including random perturbations, votings and aggregations.

Table 21: Vicuna: ASRs of JailBreak with **Swap** random smoothing on Behaviours.

| SMOOTH $q$ | 0 | 1 | 3 | 5 |
|---|---|---|---|---|
| VICUNA | 91.8 | 71.8 | 20.9 | 0.9 |
| PRO-VICUNA-HUBER $\delta = 0.1$ | 1.8 | 0.9 | 0.9 | 0.9 |
| PRO-VICUNA-HUBER $\delta = 0.2$ | 8.2 | 1.8 | 0.9 | 0.9 |
| PRO-VICUNA-HUBER $\delta = 0.3$ | 21.8 | 2.7 | 0.9 | 0.9 |
| PRO-VICUNA-HUBER $\delta = 0.4$ | 30.9 | 21.8 | 0.9 | 0.9 |
| PRO-VICUNA-HUBER $\delta = 0.5$ | 36.4 | 23.6 | 0.9 | 0.9 |
| PRO-VICUNA-HUBER $\delta = 0.8$ | 61.8 | 40.0 | 0.9 | 0.9 |
| PRO-VICUNA-HUBER $\delta = 1.0$ | 70.0 | 53.6 | 11.8 | 0.9 |
| PRO-VICUNA-HUBER $\delta = 1.5$ | 74.5 | 60.0 | 16.4 | 1.8 |
| PRO-VICUNA-HUBER $\delta = 2.0$ | 82.7 | 73.6 | 21.8 | 1.8 |
| PRO-VICUNA-HUBER $\delta = 3.0$ | 90.0 | 74.5 | 31.8 | 7.3 |

Table 22: Vicuna: ASRs of JailBreak with **Insert** random smoothing on Behaviours.

| SMOOTH $q$ | 0 | 1 | 3 | 5 | 10 | 15 |
|---|---|---|---|---|---|---|
| VICUNA | 91.8 | 79.1 | 44.5 | 10.9 | 4.5 | 1.8 |
| PRO-VICUNA-HUBER $\delta = 0.1$ | 1.8 | 0.9 | 0.9 | 0.9 | 0.9 | 0.9 |

Table 23: Vicuna: ASRs of JailBreak with **Patch** random smoothing on Behaviours.

| SMOOTH $q$ | 0 | 1 | 3 | 5 | 10 | 15 |
|---|---|---|---|---|---|---|
| VICUNA | 91.8 | 71.8 | 57.3 | 39.1 | 21.8 | 14.5 |
| PRO-VICUNA-HUBER $\delta = 0.1$ | 1.8 | 0.9 | 0.9 | 0.9 | 0.9 | 0.9 |

## I.2 Adaptive Jailbreak

Although our Pro-Vicuna has demonstrated significant effectiveness under transfer attack (black-box), it is still unclear whether our method can be resilient under white-box attacks which adaptively target the specific victim models. The comparison of our Pro-Vicuna and backbone Vicuna under adaptive jailbreak is presented in Table 24 and Figure 23. Our Pro-Vicuna can improve Vicuna by an average of $10.4\%$ across various numbers of attack queries. We don't include SmoothLLM in adaptive attacks since it introduces non-differentiable operators that preclude the gradient-based GCG attack on it.

Table 24: Vicuna: ASRs of **Adaptive** JailBreak on Behaviours

| NUM OF ATTACK QUERIES | 12 | 13 | 14 | 15 | 16 | 17 | 18 | 19 | 20 |
|---|---|---|---|---|---|---|---|---|---|
| VICUNA | 61.4 | 65.2 | 71.5 | 75.8 | 78.7 | 82.6 | 84.1 | 86.5 | 87.4 |
| PRO-VICUNA (BEST) | 50.7 | 55.9 | 60.8 | 64.3 | 67.4 | 70.5 | 74.0 | 77.7 | 78.6 |
| PRO-VICUNA-HUBER $\delta = 0.3$ | 60.2 | 60.2 | 65.0 | 70.9 | 72.8 | 75.7 | 77.7 | 77.7 | 78.6 |
| PRO-VICUNA-HUBER $\delta = 0.5$ | 60.8 | 61.8 | 62.7 | 66.7 | 67.6 | 71.6 | 78.4 | 81.4 | 82.4 |
| PRO-VICUNA-HUBER $\delta = 0.7$ | 50.7 | 55.9 | 60.8 | 64.3 | 67.4 | 70.5 | 74.0 | 79.7 | 82.4 |

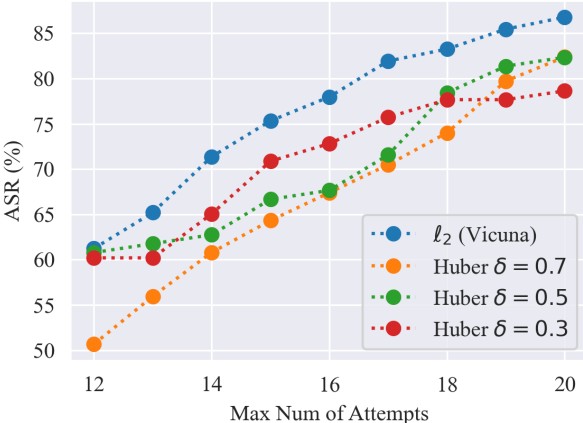

Figure 23: Adaptive JailBreak

# J  Additional Experiments on ViT

## J.1  Main Results

The main results of FGSM on ViT are presented in Table 25. We can conclude that our Pro-ViT (MCP) outperforms other methods across various budgets.

Table 25: Adversarial robustness on CIFAR-10 (FGSM).

| MODEL \ BUDGET | 0 (CLEAN) | 8/255 | 4/255 | 1/255 |
|---|---|---|---|---|
| VIT | 98.74 | 35.05 | 39.04 | 60.43 |
| PRO-VIT-L1 | 98.46 | 42.41 | 46.33 | 61.99 |
| PRO-VIT-HUBER | 98.76 | 37.47 | 41.93 | 62.72 |
| PRO-VIT-MCP (OURS) | 98.40 | 54.10 | 60.38 | 75.85 |

Table 26: Adversarial Robsutness on Imagenet-1k (PGD).

| MODEL | 0 (CLEAN) | 8/255 | 4/255 | 2/255 | 1/255 |
|---|---|---|---|---|---|
| VIT | 90.7 | 0.0 | 0.1 | 4.1 | 30.2 |
| PRO-VIT-L1 | 89.1 | 5.4 | 14.2 | 31.6 | 47.6 |
| PRO-VIT-HUBER | 90.6 | 4.3 | 15.1 | 32.7 | 51.6 |
| PRO-VIT-MCP (OURS) | 88.6 | 13.0 | 23.1 | 49.1 | 65.6 |

## J.2  Ablation study.

The ablation study of PGD and FGSM are provided in Table 27, Table 28 and Figure 24. As shown in the results, the optimal $\gamma$ of MCP fall into the range of (3,4). The robust estimators with more layers show the better robustness while slightly sacrifice the clean performance.

Table 27: Ablation: CIFAR-10 (PGD)

| MODEL \ BUDGET | 0 (CLEAN) | 8/255 | 4/255 | 1/255 |
|---|---|---|---|---|
| PRO-VIT-HUBER $K=3, \delta=1$ | 98.43 | 0.09 | 0.82 | 28.38 |
| PRO-VIT-HUBER $K=3, \delta=3$ | 98.56 | 0.07 | 1.36 | 31.04 |
| PRO-VIT-HUBER $K=3, \delta=5$ | 98.56 | 0.09 | 1.57 | 34.9 |
| PRO-VIT-HUBER $K=3, \delta=7$ | 98.76 | 0.15 | 1.72 | 34.89 |
| PRO-VIT-HUBER $K=3, \delta=9$ | 98.75 | 0.18 | 1.86 | 34.98 |
| PRO-VIT-MCP $K=1, \gamma=4$ | 98.07 | 1.03 | 2.43 | 26.16 |
| PRO-VIT-MCP $K=2, \gamma=4$ | 96.92 | 2.22 | 3.85 | 39.04 |
| PRO-VIT-MCP $K=3, \gamma=4$ | 95.79 | 6.47 | 12.65 | 65.15 |
| PRO-VIT-MCP $K=4, \gamma=4$ | 94.29 | 14.64 | 27.17 | 74.72 |
| PRO-VIT-MCP $K=5, \gamma=4$ | 93.43 | 23.34 | 37.56 | 76.75 |
| PRO-VIT-MCP $K=6, \gamma=4$ | 92.56 | 28.94 | 43.34 | 77.39 |
| PRO-VIT-MCP $K=7, \gamma=4$ | 91.89 | 31.57 | 47.01 | 76.86 |
| PRO-VIT-MCP $K=8, \gamma=4$ | 91.36 | 33.4 | 47.22 | 76.16 |
| PRO-VIT-MCP $K=9, \gamma=4$ | 90.76 | 33.17 | 48.11 | 75.56 |
| PRO-VIT-MCP $K=3, \gamma=2$ | 98.4 | 2.95 | 6.19 | 56.41 |
| PRO-VIT-MCP $K=3, \gamma=3$ | 97.97 | 5.8 | 10.83 | 67.07 |
| PRO-VIT-MCP $K=3, \gamma=4$ | 95.79 | 6.47 | 12.65 | 65.15 |
| PRO-VIT-MCP $K=3, \gamma=5$ | 92.77 | 3.53 | 7.54 | 45.70 |
| PRO-VIT-MCP $K=3, \gamma=6$ | 94.0 | 3.54 | 7.22 | 37.22 |

Table 28: Ablation: CIFAR-10 (FGSM)

| Model \ Budget | 0 (Clean) | 8/255 | 4/255 | 1/255 |
|---|---|---|---|---|
| Pro-ViT-Huber $K = 3, \delta = 1$ | 98.43 | 36.23 | 40.57 | 58.84 |
| Pro-ViT-Huber $K = 3, \delta = 3$ | 98.56 | 36.99 | 40.88 | 60.54 |
| Pro-ViT-Huber $K = 3, \delta = 5$ | 98.65 | 37.47 | 41.93 | 62.72 |
| Pro-ViT-Huber $K = 3, \delta = 7$ | 98.76 | 36.55 | 41.02 | 62.20 |
| Pro-ViT-Huber $K = 3, \delta = 9$ | 98.75 | 35.75 | 40.39 | 61.38 |
| Pro-ViT-MCP $K = 1, \gamma = 4$ | 98.07 | 35.22 | 38.12 | 52.82 |
| Pro-ViT-MCP $K = 2, \gamma = 4$ | 96.92 | 39.84 | 43.19 | 55.49 |
| Pro-ViT-MCP $K = 3, \gamma = 4$ | 95.79 | 47.38 | 53.6 | 67.03 |
| Pro-ViT-MCP $K = 4, \gamma = 4$ | 94.29 | 49.26 | 58.49 | 72.71 |
| Pro-ViT-MCP $K = 5, \gamma = 4$ | 93.42 | 49.42 | 59.03 | 74.35 |
| Pro-ViT-MCP $K = 6, \gamma = 4$ | 92.56 | 48.23 | 59.21 | 76.01 |
| Pro-ViT-MCP $K = 3, \gamma = 2$ | 98.4 | 47.98 | 52.39 | 70.59 |
| Pro-ViT-MCP $K = 3, \gamma = 3$ | 97.97 | 51.64 | 57.21 | 73.16 |
| Pro-ViT-MCP $K = 3, \gamma = 4$ | 95.79 | 47.38 | 53.6 | 67.03 |
| Pro-ViT-MCP $K = 3, \gamma = 5$ | 92.77 | 35.37 | 41.76 | 52.10 |
| Pro-ViT-MCP $K = 3, \gamma = 6$ | 94.0 | 37.08 | 41.42 | 48.56 |
| Pro-ViT-MCP $K = 3, \gamma = 3$ | 97.97 | 51.64 | 57.21 | 73.16 |
| Pro-ViT-MCP $K = 4, \gamma = 3$ | 97.76 | 54.1 | 59.66 | 75.30 |
| Pro-ViT-MCP $K = 5, \gamma = 3$ | 97.75 | 53.29 | 60.08 | 75.85 |
| Pro-ViT-MCP $K = 6, \gamma = 3$ | 97.74 | 52.37 | 60.38 | 75.70 |

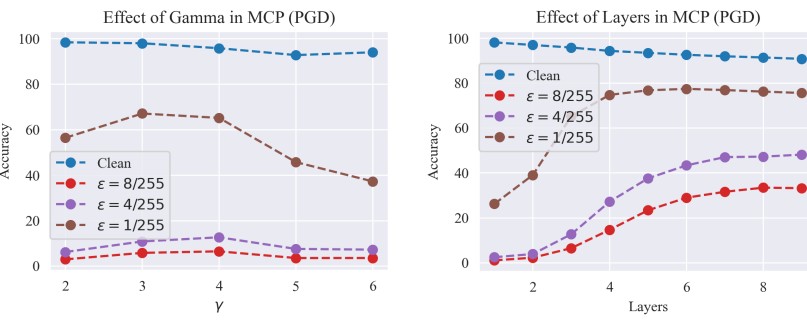

(a) Ablation study on $\gamma$ in MCP (PGD)  (b) Ablation study on $K$ in MCP (PGD)

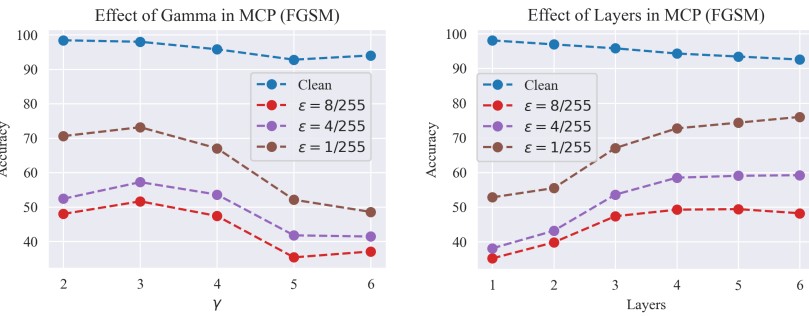

(c) Ablation study on $\gamma$ in MCP (FGSM)  (d) Ablation study on $L$ in MCP (FGSM)

Figure 24: Ablation study in MCP

# K Additional Experiments in GAT

The main results on Citeseer and the ablation study on Cora-ML are presented in Table 29 and Table 30. From the results, we can make the following conclusions: (1) Our Pro-GAT outperform other methods significantly. Under the larger budgets, the outliers introduced by the adversarial attack

will enlarge the bias effect of the estimation. In this scenario, our MCP function can mitigate or even remove the effect of outlying values in the large-value region. (2) The parameter $\gamma$ provide an implication for the robustness. For small budget, the models with large $\gamma$ perform well since it is more similar the original attention. While for large budget, the models with small $\gamma$ offer better robustness cause it can mitigate the bias introduced by the outliers.

Table 29: Results on Citeseer.

| MODEL \ BUDGET | 0% | 5% | 10% | 20% | 30% | 40% |
|---|---|---|---|---|---|---|
| GCN | $\mathbf{74.8 \pm 1.2}$ | $66.1 \pm 1.0$ | $60.9 \pm 0.8$ | $53.0 \pm 1.0$ | $47.0 \pm 0.8$ | $41.2 \pm 1.1$ |
| GNNGUARD | $72.4 \pm 1.1$ | $65.6 \pm 0.9$ | $61.8 \pm 1.4$ | $55.6 \pm 1.4$ | $51.0 \pm 1.3$ | $47.3 \pm 1.3$ |
| RGCN | $74.4 \pm 1.0$ | $66.0 \pm 0.8$ | $60.6 \pm 0.9$ | $52.5 \pm 0.8$ | $46.1 \pm 0.9$ | $40.2 \pm 1.0$ |
| GRAND | $\mathbf{74.8 \pm 0.6}$ | $66.6 \pm 0.7$ | $61.8 \pm 0.7$ | $53.6 \pm 1.1$ | $47.4 \pm 1.2$ | $42.2 \pm 0.9$ |
| PROGNN | $74.2 \pm 1.3$ | $65.6 \pm 1.1$ | $60.3 \pm 1.1$ | $52.7 \pm 1.4$ | $46.2 \pm 0.9$ | $40.8 \pm 0.6$ |
| JACCARD-GCN | $\mathbf{74.8 \pm 1.2}$ | $66.3 \pm 1.2$ | $60.9 \pm 1.2$ | $53.3 \pm 0.9$ | $46.5 \pm 0.9$ | $41.1 \pm 1.0$ |
| SOFTMEDIAN | $74.6 \pm 0.7$ | $68.0 \pm 0.7$ | $64.4 \pm 0.9$ | $59.3 \pm 1.1$ | $55.2 \pm 2.0$ | $51.9 \pm 2.1$ |
| GAT | $73.4 \pm 1.2$ | $65.4 \pm 1.3$ | $60.4 \pm 1.4$ | $52.6 \pm 2.5$ | $47.2 \pm 3.4$ | $41.2 \pm 4.8$ |
| PRO-GAT (OURS) | $73.4 \pm 1.1$ | $\mathbf{68.9 \pm 1.4}$ | $\mathbf{66.0 \pm 2.2}$ | $\mathbf{63.0 \pm 2.4}$ | $\mathbf{59.5 \pm 2.6}$ | $\mathbf{57.7 \pm 2.0}$ |

Table 30: Ablation study on Cora-ML.

| MODEL \ BUDGET | 0% (CLEAN) | 5% | 10% | 20% | 30% | 40% |
|---|---|---|---|---|---|---|
| $K = 1, \gamma = 1.0$ | $84.14 \pm 0.35$ | $78.51 \pm 0.39$ | $75.70 \pm 0.45$ | $72.06 \pm 0.44$ | $69.00 \pm 0.65$ | $66.34 \pm 0.99$ |
| $K = 1, \gamma = 2.0$ | $83.70 \pm 0.72$ | $78.46 \pm 0.51$ | $75.46 \pm 0.80$ | $71.21 \pm 1.32$ | $68.39 \pm 1.60$ | $65.91 \pm 2.17$ |
| $K = 1, \gamma = 3.0$ | $83.95 \pm 0.77$ | $77.93 \pm 0.55$ | $74.35 \pm 0.63$ | $69.46 \pm 1.16$ | $66.31 \pm 1.73$ | $62.87 \pm 1.54$ |
| $K = 1, \gamma = 4.0$ | $84.18 \pm 0.64$ | $77.41 \pm 0.64$ | $73.97 \pm 0.74$ | $68.97 \pm 0.98$ | $65.70 \pm 1.20$ | $62.70 \pm 1.42$ |
| $K = 1, \gamma = 5.0$ | $83.91 \pm 1.17$ | $77.57 \pm 0.93$ | $73.88 \pm 1.29$ | $68.98 \pm 1.27$ | $65.17 \pm 1.54$ | $62.40 \pm 1.75$ |
| $K = 1, \gamma = 6.0$ | $83.91 \pm 0.79$ | $77.45 \pm 0.75$ | $73.56 \pm 0.88$ | $68.66 \pm 1.21$ | $64.80 \pm 1.41$ | $61.94 \pm 2.08$ |
| $K = 1, \gamma = 7.0$ | $84.26 \pm 0.54$ | $77.77 \pm 0.79$ | $74.21 \pm 0.67$ | $68.94 \pm 0.88$ | $65.20 \pm 1.30$ | $62.31 \pm 1.69$ |
| $K = 3, \gamma = 1.0$ | $82.75 \pm 0.87$ | $77.59 \pm 0.95$ | $75.04 \pm 1.25$ | $71.47 \pm 1.06$ | $68.70 \pm 1.20$ | $66.53 \pm 1.24$ |
| $K = 3, \gamma = 2.0$ | $80.88 \pm 3.79$ | $75.89 \pm 2.88$ | $72.61 \pm 2.39$ | $68.71 \pm 2.07$ | $65.39 \pm 2.25$ | $62.29 \pm 2.65$ |
| $K = 3, \gamma = 3.0$ | $83.04 \pm 1.04$ | $77.09 \pm 1.22$ | $73.82 \pm 1.23$ | $69.27 \pm 1.45$ | $65.71 \pm 1.62$ | $62.62 \pm 2.07$ |
| $K = 3, \gamma = 4.0$ | $81.84 \pm 3.57$ | $76.37 \pm 2.62$ | $73.45 \pm 2.04$ | $68.63 \pm 2.49$ | $65.09 \pm 2.56$ | $62.42 \pm 2.33$ |
| $K = 3, \gamma = 5.0$ | $83.79 \pm 0.75$ | $77.81 \pm 0.85$ | $74.58 \pm 0.96$ | $69.90 \pm 1.05$ | $66.32 \pm 1.26$ | $63.33 \pm 1.66$ |
| $K = 3, \gamma = 6.0$ | $83.38 \pm 1.12$ | $77.17 \pm 1.15$ | $74.12 \pm 1.28$ | $69.27 \pm 1.33$ | $65.59 \pm 1.34$ | $62.86 \pm 1.75$ |
| $K = 3, \gamma = 7.0$ | $84.57 \pm 0.76$ | $78.47 \pm 0.78$ | $75.15 \pm 0.84$ | $70.47 \pm 0.96$ | $66.91 \pm 1.33$ | $63.94 \pm 1.33$ |

## L    Complexity Analysis.

Here we will provide a complexity analysis of the vanilla attention, our robust attention, KDE-based attention and RKDE-based attention. The related notations are $\mathbf{Q}, \mathbf{K}, \mathbf{V} \in \mathbb{R}^{N \times D}$ and $\mathbf{A} \in \mathbb{R}^{N \times N}$. The major difference in these methods is how to derive the attention matrix, therefore we will only count the complexity of attention matrix derivation and context matrix computation.

- **Vanilla Attention.** The vanilla attention matrix in can be formulated as $\mathbf{A} = \text{softmax}\left(\frac{\mathbf{Q}\mathbf{K}^{\top}}{\sqrt{D}}\right)$, which costs about $N \times N \times D$. The context matrix computation requires $N \times N \times D$, so the total cost is $2 \cdot (N \times N \times D)$

- **Our ProAttention.** For our robust attention, we need to firstly compute the original matrix $\mathbf{A}$ ($N \times N \times D$). Then we need to compute weight $\mathbf{W}^{(k)}$ based on the pairwise distance between the $\mathbf{V}$ and current estimator $\mathbf{Z}^{(k)}$ ($N \times N \times D$). Finally we need to update the estimator by $\mathbf{Z}^{(k+1)} = (\mathbf{A} \odot \mathbf{W}^{(k)})\mathbf{V}$ ($N \times N \times D$). The context matrix is calculated through the iterations. Therefore, the total cost will be $(1 + 2K) \cdot N \times N \times D$. As stated in the ablation study in Section 4.2.2, our Newton-IRLS in ProAttention can converge efficiently within 3 steps ($K \leq 3$). Therefore, our ProAttention is still effeicient.

  **Kernel Density Estimation (KDE) Attention.** For KDE-based attention, the attention matrix is computed based on the pairwise ditance between the $\mathbf{K}$ and $\mathbf{Q}$, which cost $N \times N \times D$. The context matrix computation requires $N \times N \times D$, so the total cost is $2 \cdot (N \times N \times D)$.

- **Robust Kernel Density Estimation (RKDE) Attention.** For the RKDE-based attention, we need to perform the following operations. Firstly, we need to compute the basic matrix ad KDE attention matrix $\mathbf{A}$ ($N \times N \times D$). Then we need to compute pairwise distance $\mathbf{D}_{\mathbf{K}}^{(k)}$ for all the key pairs ($N \times N \times D$) and update the weight $\mathbf{W}_{\mathbf{K}}^{(k)}$ based on $\mathbf{W}_{\mathbf{K}}^{(k-1)}$ and $\mathbf{D}_{\mathbf{K}}^{(k)}$ ($N \times N \times N$). Similarly, we need calculate the pairwise distance $\mathbf{D}_{\mathbf{KV}}^{(k)}$ and update the weight $\mathbf{W}_{\mathbf{KV}}^{(k)}$ for concatenated key and value, which costs $N \times N \times 2D$ and $N \times N \times N$, repectively. The context matrix computation requires $N \times N \times D$. Therefore, the total cost will be $(2 + 3K) \cdot N \times N \times D + 2K \cdot N \times N \times N$.

