# OpenReview forum: "ProTransformer: Robustify Transformers via Plug-and-Play Paradigm"
_NeurIPS.cc/2024/Conference — NeurIPS 2024 poster_

### Official Review · Reviewer_Yhky · 2024-07-08

**Soundness:** 3
**Presentation:** 2
**Contribution:** 3
**Rating:** 5
**Confidence:** 4

**Summary:**

This paper proposes a robust attention mechanism to improve the resilience of transformer-based architectures. The method does not need additional training or finetuning with only four lines of code, which is simple yet effective. To show the effectiveness of the proposed mechanism, the paper conducts experiments across different tasks including topic classification, sentiment analysis, textual entailment, jailbreak attack, on different backbone architectures such as large language models. The results show promising resilience.

**Strengths:**

1. The method is simple yet effective. With only four lines of codes, it's able to provide good resilience performance across diverse tasks and model architectures.
2. The experiments are quite comprehensive. The study of robust transfomers include language modeling, image classification, and graph representative learning. For language modeling, the evaluations consider classic text attacks, prompt attacks, and jailbreak attacks.
3. The results demonstrate promising performance.
4. The plug-and-play nature of ProTransformer makes it practical for real-world applications, as it can be easily integrated into existing model architectures without significant overhead.

**Weaknesses:**

1. As the paper points about one of the advantages of the method is the efficiency (efficient Newton-ISLR algorithm), there should be more discussions about the runtime of the proposed method across different architectures. The current version only includes the study in Tab.2 on a single model type. What is the influence on LLMs as they are more computation intensive? And how is the runtime compared with other robust designs?

2. For the results on image classification tasks, there is no comparision. How is the performance of the proposed method compared with other robust designs such as [*]?

[*] Robustifying Token Attention for Vision Transformers

**Questions:**

1. For the main results in Tab.1, how many layers (K) in the transformer models are equipped with ProAttention?
2. For the results in Tab.2, what is the robutness performance by employing different numbers of ProAttention layers?

**Limitations:**

Please refer to weakness and questions.

---

> ### Author Rebuttal · Authors · 2024-08-07
>
> Thanks for your recognition of the efficiency and effectiveness of our method. We are glad to solve134
> your concerns and answer your questions with the following illustrations.
>
> **W1**: As the paper points about one of the advantages of the method is the efficiency (efficient Newton-ISLR algorithm), there should be more discussions about the runtime of the proposed method across different architectures. The current version only includes the study in Table 2 on a single model type. What is the influence on LLMs as they are more computation intensive? And how is the runtime compared with other robust designs?
>
> **Answer.** I include the average running time across different architectures. The results show the consistent conclusion with the BERT backbone: Our ProTransformer will only include 1-2 times of inference time while achieving great robustness.
> We want to point out that robustifying models is usually very time-consuming and labor-intensive. For example, adversarial training methods usually require substantial training time (e.g., 7-100 times of the normal training time [1]), while our ProTransformer can directly plug in the robust attention into a pre-trained backbone model without any training. Moreover, randomized smoothing methods usually require a lot of inference time (e.g., 1,000 times for the language domain [2] and 100,000 times for the vision domain [3]). Additionally, most robust architectures are quite complicated, require training the models from scratch, and exhibit limited robustness. Therefore, our ProTransformer, which only requires 1-2 times more inference time, is quite efficient and economical.
>
> [1] Aleksander Madry, Aleksandar Makelov, Ludwig Schmidt, Dimitris Tsipras, and Adrian Vladu. "Towards deep learning models resistant to adversarial attacks." In International Conference on Learning Representations, 2018
>
> [2] Lou, Qian, et al. "CR-UTP: Certified Robustness against Universal Text Perturbations." arXiv preprint arXiv:2406.01873 (2024).
>
> [3] Cohen, Jeremy, Elan Rosenfeld, and Zico Kolter. "Certified adversarial robustness via randomized smoothing." International conference on machine learning. PMLR, 2019.
>
> [4] Han, Xing, et al. "Designing robust transformers using robust kernel density estimation." Advances in Neural Information Processing Systems 36 (2024).
>
> ### Table 5: Average running time (ms) on AG-NEWS.
>
> | # Layers (K) | 0 (vanilla) | 1     | 2     | 3     | 4     | 5     | 6     |
> |--------------|-------------|-------|-------|-------|-------|-------|-------|
> | ProBERT      | 6.14        | 9.04  | 11.67 | 14.34 | 17.33 | 19.89 | 21.87 |
> | ProALBERT    | 5.63        | 8.70  | 13.03 | 16.10 | 19.37 | 22.77 | 23.27 |
> | ProDistilBERT| 2.17        | 4.16  | 6.88  | 8.38  | 9.17  | 11.95 | 13.19 |
> | ProRoberta   | 4.95        | 3.10  | 11.86 | 14.93 | 18.40 | 20.89 | 23.04 |
>
> **W2**: For the results on image classification tasks, there is no comparison. How is the performance of the proposed method compared with other robust designs such as [*]?
>
> [*] Robustifying Token Attention for Vision Transformers
>
> **Answer.** We validate the advantage of our ProTransformer over RVT [1] and TAP&ADL [2] in the following table.
>
> ### Table 6: Adversarial robustness under PGD.
>
> | Model               | Clean  | PGD    |
> |---------------------|--------|--------|
> | ViT                 | 98.74  | 0.26   |
> | RVT [1]             | 97.65  | 18.71  |
> | FAN-B-Hybrid +TAP&ADL [2] | 98.45  | 22.23  |
> | Pro-ViT (Ours)      | 98.40  | 33.40  |
>
> [1] Towards Robust Vision Transformer
>
> [2] Robustifying Token Attention for Vision Transformers
>
> **Q1**: For the main results in Tab.1, how many layers (K) in the transformer models are equipped with ProAttention?
>
> **Answer.** We equipped ProAttention with 3 layers (K).
>
> **Q2**: For the results in Tab.2, what is the robustness performance by employing different numbers of ProAttention layers?
>
> **Answer**: We already include the ablation study on the number of layers in the Appendix G.6.2. The results validate that our algorithm can converge well within 3 steps.
>
> | Model               | Clean | Textfooler |
> |---------------------|-------|------------|
> | Pro-BERT-MCP K = 5  | 93.6  | 39.2       |
> | Pro-BERT-MCP K = 4  | 93.7  | 37.2       |
> | Pro-BERT-MCP K = 3  | 93.2  | 37.8       |
> | Pro-BERT-MCP K = 2  | 93.7  | 33.9       |
> | Pro-BERT-MCP K = 1  | 93.9  | 26.8       |

---

> > ### Comment · Reviewer_Yhky · 2024-08-13
> >
> > Thanks for the response from the authors. The response have answered my questions and I will maintain my rating of acceptance  for the paper.

---

### Official Review · Reviewer_h5FP · 2024-07-11

**Soundness:** 3
**Presentation:** 3
**Contribution:** 3
**Rating:** 6
**Confidence:** 4

**Summary:**

This paper proposed a robust transformer architecture named ProTransformer through a plug-and-play paradigm without further training or fine-tuning. The authors include robust token estimators in the self-attention blocks which are more resilient to the dominating impact of input tokens, and apply Newton-ISLR to approximate these estimators. The comprehensive experiments for various tasks, modalities, attacks, and backbones show that ProTransformer can significantly improve the robustness of transformers without compromising benign performance.

**Strengths:**

1. This paper is well-written with clear descriptions of their method with both motivation/intuition analysis and theoretical analysis.
2. Comprehensive experiments are conducted to prove their claims of the enhanced robustness of Transformers. This includes robustness analysis for both the traditional adversarial attacks within the word or character level and the recent jailbreak attacks for large language models. Detailed ablation studies and transformers for alternative modalities are also considered.
3. The ProTransformer is simple to use (Plug-and-Play) and has significantly improved the robustness. Moreover, ProTransformers can not only work on language modeling but also show its generalization for transformer-based architectures in vision and graph domains.

**Weaknesses:**

1. Lack of analysis of clean performance in jailbreak attacks. I want to know if ProTransformer would hurt the generation quality of LLMs.
2. Less discussion about the semantic level attacks and robustness of corruption. For example, the AdvGlue Benchmark (https://adversarialglue.github.io/) for corruption robustness and PAIR[1] for semantic-level jailbreak attacks.
3. Lack of comparison with baseline defense methods.

[1] Chao, Patrick, Alexander Robey, Edgar Dobriban, Hamed Hassani, George J. Pappas, and Eric Wong. Jailbreaking black box large language models in twenty queries. arXiv preprint arXiv:2310.08419 (2023).

**Questions:**

1. When using white-box adaptive attacks like PGD and GCG, do you compute the gradient through your ProAttention blocks?
2. For the jailbreak attack results in Figure 6, why do you use Huber instead of MCP attack constraints? As shown in Section 4.2.1, MCP shows the best performance.

**Limitations:**

The limitations of the paper are not adequately discussed. The cost of the evaluation with ProTransformer should be included in the limitations.

---

> ### Author Rebuttal · Authors · 2024-08-07
>
> Thanks for your recognition of the novelty and effectiveness of our method. We are glad to solve33
> your concerns and answer your questions with the following illustrations.
>
> **W1**: Lack of analysis of clean performance in jailbreak attacks. I want to know if ProTransformer would hurt the generation quality of LLMs.
>
> **Answer.** In ProTransformer, using Huber loss will almost not hurt the generation quality while using L1 or MCP will slightly sacrifice the generation performance. We construct 100 synthetic texts (including prompts and reference texts) evaluate the generation with the BLEU score and a series of ROUGE scores with the vicuna-7b as the backbone. The following results verify that Huber model keep similar score with L2 (vanilla), while L1 and MCP may sacrifice a little bit quality.
>
> ### Table 2: BLEU and ROUGE score on generation quality.
>
> | Penalty in ProTransformer | L2 (Vanilla) | Huber | L1   | MCP   |
> |----------------------------|--------------|-------|------|-------|
> | BLEU                       | 0.151        | 0.154 | 0.145| 0.136 |
> | ROUGE-1                    | 0.385        | 0.390 | 0.368| 0.375 |
> | ROUGE-2                    | 0.379        | 0.353 | 0.344| 0.314 |
> | ROUGE-L                    | 0.396        | 0.390 | 0.385| 0.368 |
>
> **W2**: Less discussion about the semantic level attacks and robustness of corruption. For example, the AdvGlue Benchmark ([https://adversarialglue.github.io/](https://adversarialglue.github.io/)) for corruption robustness and PAIR[1] for semantic-level jailbreak attacks.
>
> [1] Chao, Patrick, Alexander Robey, Edgar Dobriban, Hamed Hassani, George J. Pappas, and Eric Wong. "Jailbreaking black box large language models in twenty queries." _arXiv preprint arXiv:2310.08419 (2023)_.
>
> **Answer.** For AdvGlue Benchmark, we already included the evaluation under typos-based Textbugger and embedding similarity-based Textfooler. We add the experiment of the context-aware BERT-Attack in the following table, the results show our Pro-BERT outperforms all of the other baselines.
>
> ### Table 3: ASR (%) under BERT-Attack.
>
> | Method  | Aua% | ASR% |
> |---------|------|------|
> | BERT    | 5.8  | 93.7 |
> | FreeLB  | 21.7 | 76.6 |
> | PGD     | 21.0 | 77.6 |
> | MixADA  | 7.6  | 91.8 |
> | TA-VAT  | 19.2 | 79.4 |
> | SAFER   | 38.5 | 58.8 |
> | RanMASK | 36.0 | 58.4 |
> | Pro-BERT (Ours) | 42.7 | 54.4 |
>
> Additionally, we also evaluate the advantage of our ProTransformer under semantic-level jailbreak attack PAIR in the following table.
>
> ### Table 4: ASR (%) under PAIR Jailbreaks.
>
> | Model                    | ASR(%) |
> |--------------------------|--------|
> | Vicuna                   | 98.7   |
> | Vicuna+SmoothLLM         | 51.4   |
> | Pro-Vicuna               | 48.9   |
> | Pro-Vicuna+SmoothLLM     | 41.6   |
>
> **W3**: Lack of comparison with baseline defense methods.
>
> **Answer.** Actually we have included the comparison with baseline defense methods for each domains:
> - Language domain: backbone baselines ALBERT, DistilBERT, RoBERTa and BERT, and popular defenses including MixADA, PGD-Adv, FreeLB, TA-VAT and SmoothLLM.
> - Vision domain: We include Deit, Convit, BeiT, Swin, ViT.
> - Graph domain: GCN, GAT, GNNGuard, RGCN, GRAND, ProGNN, Jaccard-GCN and Soft-Median.
>
> **Q1**: When using white-box adaptive attacks like PGD and GCG, do you compute the gradient through your ProAttention blocks?
> **Answer.** Yes. We compute the gradient through the ProAttention blocks.
>
> **Q2**: For the jailbreak attack results in Figure 6, why do you use Huber instead of MCP attack constraints? As shown in Section 4.2.1, MCP shows the best performance.
>
> **Answer.** In Section 4.2.1, the task is classification, so both Huber and MCP losses would not sacrifice too much clean performance. But in jailbreak, we evaluate the model with the generated response. In this scenario, Huber loss has $L_2$ penalty in the low-value region, which enables the model to keep better generation quality. But MCP will hurt the generation quality, result in some non-semantic answer.
>
> **L1**: The limitations of the paper are not adequately discussed. The cost of the evaluation with ProTransformer should be included in the limitations.
>
> **Answer.** Please refer to the **global response for the concern of limitations of ProTransformer.**

---

> > ### Comment · Reviewer_h5FP · 2024-08-13
> >
> > Thanks for your response. I would maintain my score.

---

### Official Review · Reviewer_JZTY · 2024-07-12

**Soundness:** 2
**Presentation:** 2
**Contribution:** 2
**Rating:** 4
**Confidence:** 1

**Summary:**

This paper proposes an interpretable robust attention layer to robustify transformer architecture via a plug-and-play paradigm.

**Strengths:**

The proposed method is practical and can be plugged into the given transformer as a plug-and-play layer.
The experiments are robust and conclusive, signifying the efficacy and satisfactory performance of the proposed method.

**Weaknesses:**

It seems that the complexity of the proposed ProAttention is still greater than that of linear attention [1] [2] [3].

[1] Han, Dongchen, et al. "Flatten transformer: Vision transformer using focused linear attention." Proceedings of the IEEE/CVF international conference on computer vision. 2023.
[2] Katharopoulos, Angelos, et al. "Transformers are rnns: Fast autoregressive transformers with linear attention." International conference on machine learning. PMLR, 2020.
[3] Zhu, Lianghui, et al. "DiG: Scalable and Efficient Diffusion Models with Gated Linear Attention." arXiv preprint arXiv:2405.18428 (2024).

**Questions:**

1. Could the authors compare the proposed ProAttention with linear attention under the same experimental setting?

**Limitations:**

The comparison with different attention mechanisms is missing.

---

> ### Author Rebuttal · Authors · 2024-08-07
>
> Thanks for your recognition of the efficiency and effectiveness of our method. We are glad to solve134
> your concerns and answer your questions with the following illustrations.
>
> **W1**: It seems that the complexity of the proposed ProAttention is still greater than that of linear attention [1] [2] [3].
>
> [1] Han, Dongchen, et al. "Flatten transformer: Vision transformer using focused linear attention." Proceedings of the IEEE/CVF International Conference on Computer Vision. 2023.
>
> [2] Katharopoulos, Angelos, et al. "Transformers are rnns: Fast autoregressive transformers with linear attention." International conference on machine learning. PMLR, 2020.
>
> [3] Zhu, Liangxiu, et al. "DiG: Scalable and Efficient Diffusion Models with Gated Linear Attention." arXiv preprint arXiv:2405.18428 (2024).
>
> **Answer**:
> Thanks for your comments. We would like to emphasize that while we aim to develop algorithm with acceptable efficiency, efficiency is not the focus of this work. Therefore, linear attention is irrelevant to this context. Instead, our ProAttention targets on improve the robustness of transformers. Linear attention aims to improve the efficiency of transformers but is still vulnerable against adversarial attacks as verified in next answer. Their contributions are totally orthogonal.
>
> **Q1**: Could the authors compare the proposed ProAttention with linear attention under the same experimental setting?
>
> **L1**: The comparison with different attention mechanisms is missing.
>
> **Answer:**
> Thanks for your comments. We compare different attention mechanisms (vanilla attention, Linear attention [1], Longformer attention [2], and our ProAttention) on IMDB across various attacks in the following table. Our method shows better robustness compared to other attention mechanisms including the linear one.
>
> [1] Katharopoulos, Angelos, et al. "Transformers are rnns: Fast autoregressive transformers with linear attention." International conference on machine learning. PMLR, 2020.
>
> [2] Beltagy, Iz, Matthew E. Peters, and Arman Cohan. "Longformer: The long-document transformer." arXiv preprint arXiv:2004.05150 (2020).
>
>
> ### Table 1: The results of sentiment analysis on IMDB.
>
> | Model             | Clean | Textfooler | TextBugger | DeepWordBug | PWWS |
> |-------------------|-------|------------|------------|-------------|------|
> | Vanilla attention | 92.3  | 11.8       | 11.3       | 32.8        | 26.4 |
> | Linear attention  | 89.4  | 6.9        | 5.4        | 20.7        | 17.7 |
> | Longformer attention | 87.5  | 6.4        | 7.9        | 28.5        | 23.2 |
> | ProAttention (L1) (Ours) | 93.3  | 24.6       | 13.0       | 36.0        | 32.7 |
> | ProAttention (Huber) (Ours) | 93.0  | 24.8       | 13.4       | 36.9        | 31.5 |
> | ProAttention (MCP) (Ours) | 93.5  | 22.1       | 44.6       | 55.5        | 56.3 |

---

### Official Review · Reviewer_nL5G · 2024-07-13

**Soundness:** 3
**Presentation:** 3
**Contribution:** 3
**Rating:** 6
**Confidence:** 4

**Summary:**

In this paper, the authors intend to robustify transformer architectures against adversarial attacks to enhance their resilience across various machine learning tasks. Specifically, they propose the ProAttention mechanism. They use a novel interpretation of the self-attention mechanism as a weighted least squares (WLS) estimator, along with robust WLS token estimators and an efficient Newton-IRLS algorithm with convergence guarantees, which enhances the robustness of transformers without additional training or fine-tuning. Further, the authors develop a plug-and-play layer, ProAttention, to be integrated into existing transformers. They demonstrate several experiments to show the performance of this architecture in improving robustness across language modeling, image classification, and graph representation learning.

**Strengths:**

1. The idea of ProAttention is novel. Most existing work focuses on either architecture-agnostic defenses or robustness improvements through adversarial training, but these approaches often fail to generalize across different tasks or require substantial computational resources. In this paper, the authors propose a plug-and-play paradigm to enhance robustness against adversarial attacks without additional training or fine-tuning.

2. The ProAttention framework is reasonable, and the adaptation of the Newton-IRLS algorithm makes it efficient and effective. The framework leverages the robust weighted least squares estimator to mitigate the effects of adversarial attacks.

3. The experiments are extensive, and sufficient hyper-parameters are provided for reproduction. The authors detail their experimental setup and provide comprehensive ablation studies to demonstrate the robustness and efficiency of their approach under various attack scenarios.

**Weaknesses:**

1. The paper does not include error margins for experiments, especially for ablation studies in Section 4. The paper provides extensive experimental results; however, it does not include error bars or confidence intervals to indicate the variability or statistical significance of the results.

2. This paper could benefit from better presentation and clearer structure. For example, the captions for the figures, especially Fig. 1 and 2, are too simple and lack the necessary explanations. This may lead to misunderstandings about the designs or results.

3. I do not see a discussion of the method's limitations, despite the title of the final section mentioning “Limitation.”

4. Although the paper proposes enhancing transformer robustness through the ProAttention mechanism, it needs a thorough explanation and analysis of why this method performs better under various attacks. I suggest that the authors further discuss this aspect in more detail

**Questions:**

1. The paper claims that the Newton-IRLS algorithm converges efficiently within three steps. However, can the authors provide a more detailed analysis or proof of this convergence rate, especially for larger models or more complex datasets?

2. The authors show the effectiveness of ProAttention in enhancing robustness across language, vision, and graph domains. However, each domain's specific implementation details and hyper-parameter tuning strategies seem different. Can the authors clarify how the ProAttention framework can be uniformly applied across these diverse domains without extensive domain-specific modifications?

3. The ProAttention framework relies on the weighted least squares (WLS) estimator and its robust variants. Can authors provide a deeper theoretical justification for choosing WLS and its robustness under adversarial attacks? Specifically, how do the assumptions of WLS hold up under different types of adversarial perturbations, and are there scenarios where the WLS assumptions might fail, leading to reduced robustness?

**Limitations:**

Despite being designed for efficiency, the ProAttention framework introduces additional computational overhead during inference. The paper mentions that the ProAttention layer requires 1-2 times more inference time, which may limit its deployment in real-time applications or on devices with limited computational resources.

The ProAttention framework relies on WLS estimators and their robust variants. While the paper provides a theoretical justification for choosing WLS, its effectiveness under different adversarial perturbations needs further investigation. In some scenarios, the assumptions of WLS may fail and reduce robustness.

---

> ### Author Rebuttal · Authors · 2024-08-07
>
> Thanks for your recognition of the efficiency and effectiveness of our method. We are glad to solve134
> your concerns and answer your questions with the following illustrations.
>
> **W1**:  We want to clarify that the proposed ProAttention is plugged into fixed pre-trained models without finetuning, hence there is no randomness of optimization of training. Also, the attack strategies strictly follow the procedures without randomness. Besides, our evaluation method is common in robustness of language models, and consistent with the existing benchmarks [1] and text attacks works.
>
> [1] Searching for an effective defender: Benchmarking defense against adversarial word substitution, 2021.
>
> **W2**: Thanks for pointing it out. We have referred to the figures and included the explanations in the main text. We will include necessary captions for the figures in the revised version.
>
> - For Figure 1, we present three attack mechanisms to manipulate and harm the language models:
>   1. Classic text attacks modify the input content.
>   2. Prompt attacks perturb the prompt template.
>   3. Jailbreaks add adversarial non-semantic suffixes.
>
> - For Figure 2, we show the schematic overview of the ProTransformer: We design our ProTransformer by plugging ProAttention into transformers, replacing the vanilla attention mechanism without additional training. Furthermore, our ProTransformer is versatile and can be applied across various domains, including language, image, and graph processing.
>
> **W3**:  Please refer to the **global response for the concern of limitations of ProTransformer**.
>
> **W4**:  We have provided the detailed motivations and explanations for the robustness in Section 3. We are glad to summarize them again here:
>
> 1. In Section 3.1, we highlight the vulnerability of vanilla attention due to the quadratic impact of L2 penalty in the WLS (L2) estimator. In particular, the attacker can easily manipulate the input tokens to dominate the output because of the quadratical penalty on the difference between input and output tokens.
> 2. In Section 3.2 and 3.3, we demonstrate that robust penalties, such as L1 or MCP, can mitigate the quadratic impact to a linear or constant impact. Specifically, when some tokens are attacked and introduce large residues, our ProAttention can adaptively downweight their attention weights to mitigate their impact.
>
> To simplify the understanding, it is helpful to think about a special case of it when all the attention weight are equal: median estimation (L1 case) is notably more robust than mean estimation (quadratic case) because median estimation can be formulated as the minimizer of the sum of absolute deviations (L1 norm) which reduces the impact of outliers.
>
> **Q1**: We would like to clarify that in general, there is no way to prove that an optimization algorithm converges in three steps, especially for the challenging non-convex and non-smooth problems proposed in this paper. In addition, there is no clear connection between the convergence rate of the neural network layers and the size of models or datasets. What we provide in our paper is to theoretically guarantee each iteration step will decrease the function values so running more steps will at least not hurt. We then provide empirical evaluations to show it indeed converges fast in a few steps, which makes the algorithm practical. The exact convergence rate is neither the focus nor the primary contribution of this work, but this can be an exciting future research.
>
>
> **Q2**:
> It is common in adversarial defenses to control the balance between clean and robust performance through key hyperparameters. For instance, the famous paper TRADES [1] needs to carefully tune the combination between clean loss and robust loss to achieve best performance even for different attack budgets for the same model architectures and datasets, not to mention totally different domains evaluated in this paper. Since we plug-in our robust layers into the given models without training, it is expected some reasonable hyperparameter tuning is necessary to obtain optimal performance.
>
> We will provide some general guidance in the revision: (1) Huber loss better maintains clean performance so it typically works better when the attack is weak; (2) MCP loss works better when the attack is strong; (3) L1 loss is a special case of Huber loss; (4) the proposed Huber-MCP penalty can flexibly reduce to different options such as Huber, MCP, and L1 based on the values of hyperparameters δ and γ as shown in Figure 3 in the paper.
>
> As a future work, we can make those hyperparameters as learnable parameters learned through fine-tuning or adversarial training, which can avoid hyperparameter search but also need more training costs.
>
> [1] Zhang, Hongyang, et al. "Theoretically principled trade-off between robustness and accuracy." (ICML, 2019)
>
> **Q3**:
> We want to clarify that WLS is not an assumption; the vanilla attention is exactly equivalent to the WLS estimator (L2) under our new interpretation. Our robust ProAttention is an improved variant.
>
> **L1**: Please refer to the **global response for concern of efficiency of ProTransformer**
>
> **L2**: Please refer to Q3.

---

> > ### Comment · Reviewer_nL5G · 2024-08-11
> >
> > Thank you for your response. I look forward to seeing the updated manuscript.

---

> > > ### Author Response · Authors · 2024-08-13
> > >
> > > Thank you for your feedback. We have carefully incorporated your suggestions into our revision, but the updated manuscript cannot be uploaded at this moment. We would greatly appreciate it if you could consider our clarifications in the rebuttal.

---

### Author Rebuttal · Authors · 2024-08-07

Thanks to all the reviewers for the recognition of the novelty and effectiveness of our method. Since several reviewers have the concern about the efficiency of ProTransformer, we will elaborate the response as follows.

## **Response for the concern of efficiency of ProTransformer**

We want to point out that robustifying models is usually very time-consuming and labor-intensive. For example, adversarial training methods usually require substantial training time (e.g., 7-100 times of the normal training time [1]), while our ProTransformer can directly plug in the robust attention into a pre-trained backbone model without any training. Moreover, randomized smoothing methods usually require a lot of inference time (e.g., 1,000 times for the language domain [2] and 100,000 times for the vision domain [3]). Additionally, most robust architectures [4] are quite complicated, require training the models from scratch, and exhibit limited robustness. Therefore, our ProTransformer, which only requires 1-2 times more inference time, is quite efficient and economical. Certainly, there is still some space to further improve the efficiency of ProAttention, which can be the focus of future work.

[1] Aleksander Madry, Aleksandar Makelov, Ludwig Schmidt, Dimitris Tsipras, and Adrian Vladu. "Towards deep learning models resistant to adversarial attacks." In International Conference on Learning Representations, 2018.

[2] Lou, Qian, et al. "CR-UTP: Certified Robustness against Universal Text Perturbations." arXiv preprint arXiv:2406.01873 (2024).

[3] Cohen, Jeremy, Elan Rosenfeld, and Zico Kolter. "Certified adversarial robustness via randomized smoothing." International Conference on Machine Learning. PMLR, 2019.

[4] Han, Xing, et al. "Designing robust transformers using robust kernel density estimation." Advances in Neural Information Processing Systems 36 (2024).

## **Response for the concern of limitations of ProTransformer**

Our limitations are listed as follows:

- Despite acceptable complexity of our ProTransformer, there still exists a potential to improve the efficiency of our models.
- We majorly claims and validate the effectiveness of our model under plug-in and play paradigm. We are excited about the future of the proposed ProTransformer architecture and hope to see its full potential with training or fine-tuning on large models.

---

### Decision · Program_Chairs · 2024-09-25

**Decision:**

Accept (poster)

**Comment:**

This paper introduces a robust attention mechanism intended to enhance the robustness of transformer-based architectures against adversarial attacks. A key advantage is that it can serve as a plug-and-play layer, therefore can be easily integrated into existing transformers for enhancing robustness without additional training or fine-tuning.

Overall, all reviewers appreciate the simplicity and practicability of this method, and acknowledge that the provided experiments are comprehensive and the corresponding results are solid. Meanwhile, some concerns are also raised, mainly including: 1) some critical discussions are missing (e.g., efficiency analysis, limitation); 2) the comparisons against linear attention are missing; and 3) the presentation could be further improved.

The rebuttal is considered, which addresses most of these concerns. Consequently, three reviewers stand on the positive side of accepting this paper. The sole reviewer who raised objections, primarily concerning the comparison with linear attention, did not respond to the details provided in the rebuttal; additionally, by reading the original review and the provided rebuttal, the AC feels the rebuttal reasonably alleviates this concern. Therefore, the AC decides to ignore this negative review and agree with the other three reviewers on accepting this submission.